METHODS

# Generative Bayesian modeling to nowcast the effective reproduction number from line list data with missing symptom onset dates

**Adrian Lison** [1,2] *, **Sam Abbott** [3], **Jana Huisman** [4], **Tanja Stadler** [1,2]

**1** Department of Biosystems Science and Engineering, ETH Zurich, Zurich, Switzerland, **2** SIB Swiss Institute of Bioinformatics, Lausanne, Switzerland, **3** Centre for Mathematical Modelling of Infectious Diseases, London School of Hygiene and Tropical Medicine, London, United Kingdom, **4** Physics of Living Systems, Department of Physics, Massachusetts Institute of Technology, Cambridge, Massachusetts, United States of America

* adrian.lison@bsse.ethz.ch

**Data Availability Statement:** The simulated line list data and corresponding nowcasts, as well as all statistical models and reproducible analysis scripts are available from zenodo at https://doi.org/10.

## Abstract

The time-varying effective reproduction number $R_t$ is a widely used indicator of transmission dynamics during infectious disease outbreaks. Timely estimates of $R_t$ can be obtained from reported cases counted by their date of symptom onset, which is generally closer to the time of infection than the date of report. Case counts by date of symptom onset are typically obtained from line list data, however these data can have missing information and are subject to right truncation. Previous methods have addressed these problems independently by first imputing missing onset dates, then adjusting truncated case counts, and finally estimating the effective reproduction number. This stepwise approach makes it difficult to propagate uncertainty and can introduce subtle biases during real-time estimation due to the continued impact of assumptions made in previous steps. In this work, we integrate imputation, truncation adjustment, and $R_t$ estimation into a single generative Bayesian model, allowing direct joint inference of case counts and $R_t$ from line list data with missing symptom onset dates. We then use this framework to compare the performance of nowcasting approaches with different stepwise and generative components on synthetic line list data for multiple outbreak scenarios and across different epidemic phases. We find that under reporting delays realistic for hospitalization data (50% of reports delayed by more than a week), intermediate smoothing, as is common practice in stepwise approaches, can bias nowcasts of case counts and $R_t$, which is avoided in a joint generative approach due to shared regularization of all model components. On incomplete line list data, a fully generative approach enables the quantification of uncertainty due to missing onset dates without the need for an initial multiple imputation step. In a real-world comparison using hospitalization line list data from the COVID-19 pandemic in Switzerland, we observe the same qualitative differences between approaches. The generative modeling components developed in this work have been integrated and further extended in the R package epinowcast, providing a flexible and interpretable tool for real-time surveillance.

5281/zenodo.8279675. Real-world line list data of COVID-19 hospitalizations in Switzerland are owned by the Federal Office of Public Health (FOPH), and cannot be shared publicly due to internal data protection and privacy regulations, as they contain potentially identifying patient information. The line list data are available under terms of data protection upon request to FOPH (info@bag.admin.ch).

**Funding:** This work was supported by the ETH Zürich to AL and TS; Wellcome Trust, 200901/Z/16/Z to SA; Federal Office of Public Health (FOPH). The Federal Office of Public Health (FOPH) provided access to line list data for this study. The funders played no role in the study design, data analysis, decision to publish, or preparation of the manuscript.

**Competing interests:** I have read the journal's policy and the authors of this manuscript have the following competing interests: Tanja Stadler is president of the Swiss "Scientific Advisory Panel COVID-19" (https://science-panel-covid19.ch/).

## Author summary

During an infectious disease outbreak, public health authorities require timely indicators of transmission dynamics, such as the effective reproduction number $R_t$. Since reporting data are delayed and often incomplete, statistical methods must be employed to obtain real-time estimates of case numbers and $R_t$. Existing methods involve separate steps for imputing missing data, adjusting for reporting delays, and estimating $R_t$. This stepwise approach impedes uncertainty quantification and can lead to inconsistent smoothing assumptions across steps. In this paper, we propose an alternative approach based on generative Bayesian modeling which integrates all steps into a single nowcasting model that can be directly fit to observed data. Using synthetic and real-world line list data, we demonstrate that the generative approach better captures uncertainty and avoids bias from inconsistent assumptions. The model components of our approach have been integrated into the R package epinowcast for easy use in practice.

This is a *PLOS Computational Biology* Methods paper.

## 1 Introduction

The accurate and timely tracking of transmission dynamics is an important objective of infectious disease surveillance during epidemics. For instance, the effective reproduction number $R_t$ was used as a central indicator of transmission dynamics to guide and evaluate the public health response in many countries during the COVID-19 pandemic [1–5]. $R_t$ can be estimated from clinical case data such as confirmed cases, hospitalizations or deaths under the assumption that the proportion of infections resulting in a respective case remains constant over time [6]. Tracking $R_t$ in near real-time is challenging, however. Because of substantial delays between infection and reporting of cases, case counts by date of report only provide a delayed and potentially distorted signal of transmission dynamics [7]. Therefore, approaches to estimate $R_t$ are often based on case counts by date of symptom onset, which is ideally closer to the date of infection and independent of reporting delays [1, 8]. Such data are typically obtained from a line list, which stores individual information about each case, e. g. date of symptom onset, date of positive test, and date of report.

In this work, we address three important challenges that arise when tracking transmission dynamics from symptom onset data. First, information about the date of symptom onset is typically not available for all cases in the line list. Symptom onset information may be missing due to different reasons, including gaps in ascertainment or reporting, and asymptomatic infections [9, 10]. Since these factors can change over time, cases with missing symptom onset date cannot simply be excluded when inferring $R_t$ over time [11]. Second, in real-time surveillance, the delay between symptom onset and reporting means that cases with symptom onset close to the present may not have been reported in the line list yet, which is also known as right truncation [12]. As a result, case counts by date of symptom onset will be subject to a downward bias towards the present, and a truncation adjustment is needed to correct for this bias [13–15]. Third, when estimating $R_t$ from case counts by date of symptom onset, one needs to account for stochastic incubation periods and transmission noise to accurately relate to the time of infection [8, 16].

Previous methods have typically addressed the above challenges in three separate steps [17–19]. Specifically, cases with missing symptom onset date are first completed via an imputation

model fitted on the cases with known symptom onset date. Then, case counts by date of symptom onset are corrected for occurred-but-not-yet-reported cases via a truncation adjustment model fitted on previously observed cases and reporting delays. Finally, $R_t$ is inferred from the case counts by date of symptom onset via a non-parametric method for $R_t$ estimation that accounts for incubation periods and noise. In such a stepwise approach, estimates obtained in each step are treated as data during the subsequent step, meaning that information can only flow in one direction, such that knowledge and assumptions used in later steps cannot inform earlier steps. Moreover, the uncertainty involved in earlier steps is either neglected [17] or must be incorporated via a resampling scheme, where later steps are repeatedly applied to each sample from the current step [18]. The overall efficiency of the latter solution strongly depends on the computational effort required for downstream steps. For example, Li and White [19] combined imputation, truncation adjustment, and $R_t$ estimation in a fully Bayesian approach, which is however still stepwise internally and required the use of simple estimators for truncation adjustment and $R_t$ estimation which can be directly computed as posterior predictive quantities.

In this paper, we explore an alternative to the existing stepwise approaches by integrating missing date imputation, truncation adjustment, and effective reproduction number estimation in a single generative model [20]. In essence, we use Bayesian hierarchical modeling [21] to describe how cases arise from an infection process and are reported with stochastic, time-varying delays and potentially missing symptom onset dates. This allows us to estimate the time series of symptom onsets and the effective reproduction number over time as parameters of a single, consistent nowcasting model, where the priors and model components for imputation, truncation adjustment, and $R_t$ estimation jointly inform each other. Moreover, when estimated in a fully Bayesian framework, the uncertainty from all components is naturally represented by the posterior distribution of model parameters [22].

To compare the performance of the generative and stepwise nowcasting approaches, we study the uncertainty, over- and underprediction, and probabilistic forecast scores of nowcasts on synthetic line list data for multiple outbreak scenarios and across epidemic phases. We ensure comparability by using maximally similar modeling components in all approaches that reflect the assumptions of the synthetic data equally well. Thereby, we study sufficiently realistic nowcasting settings with time-varying reporting delays, weekday effects in reporting, and randomly missing symptom onset dates. We furthermore compare the generative and stepwise approaches on hospitalization line list data from the first and second wave of the COVID-19 epidemic in Switzerland. We find that in a stepwise approach, nowcasts can be influenced by intermediate smoothing between the steps, potentially leading to bias during phases of exponential growth or decline. In contrast, by integrating truncation adjustment with a renewal model for $R_t$ estimation, the generative approach avoids bias from intermediate smoothing and more accurately predicts exponential growth trajectories in an epidemic wave. When nowcasting from incomplete line list data, we find that generative modeling of missing symptom onset dates allows for more comprehensive uncertainty quantification. Based on these findings, we discuss the use of generative modeling approaches in real-time surveillance and implications for further model development in the future.

## 2 Methods

In the following, we describe the different stepwise and generative nowcasting approaches compared in this work. Our main nowcasting target is the so-called instantaneous effective reproduction number $R_t$, which is the expected number of secondary infections caused by a primary infection if conditions remained as on date (index) $t$. We also nowcast the number of reported cases by date of symptom onset $N_t$, which can help to identify biases in $R_t$ estimation

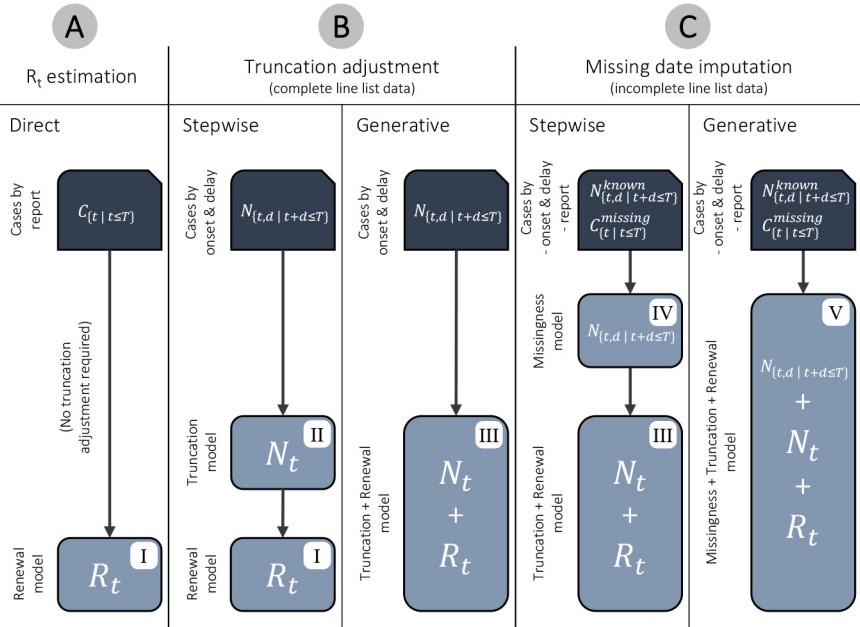

**Fig 1. Overview of different approaches for nowcasting the effective reproduction number $R_t$.** (**A**) Case counts by date of report are not biased by right truncation and can be used to estimate $R_t$ via a direct (no truncation adjustment) approach, assuming knowledge of the delay between infection and report. (**B**) Case counts by date of symptom onset are only delayed by the incubation period but subject to right truncation. The truncation can be adjusted for by using a stepwise (additional adjustment step) or generative (integrated truncation model) approach. (**C**) When line list data are incomplete, cases with missing onset date must be accounted for. This can be achieved using a stepwise (additional imputation step) or generative (integrated missingness model) approach. (**I–V**) Models used in the steps of the different approaches. To ensure comparability, the model components for $R_t$ estimation, truncation adjustment, and missing date imputation were designed to be maximally similar across all approaches (see S1 Appendix B for full model definitions).

and is often of interest by itself. The data are individual cases in a line list with a date of report and a (potentially missing) date of symptom onset. These cases can be aggregated into different counts, such as $C_t$ (number of cases reported on date $t$) or $N_{t,d}$ (number of cases with symptom onset on date $t$ and a reporting delay of $d$ days). Importantly, the counts $N_{t,d}$ are subject to right truncation, as we can only observe cases with $t + d \leq T$ until the present date $T$. Moreover, cases with missing symptom onset date can only be counted by report date, as $C_t^{\text{missing}}$. To develop our generative model for nowcasting $R_t$, we start with a model for direct $R_t$ estimation using only counts by date of report $C_t$ (Fig 1A). We then gradually increase the complexity to account for right-truncated case counts by date of symptom onset (Fig 1B) and for incomplete line list data (Fig 1C). At each stage, we first present a stepwise approach similar to existing nowcasting methods, and then show how the additional step can be directly integrated into the generative model.

## 2.1 (A) Effective reproduction number estimation

There exist various mechanistic and non-mechanistic approaches to estimate $R_t$, most of which rely on the renewal equation [23, 24] as a mathematical foundation. For example, the non-parametric method by Cori et al. [8] implemented in the widely used package EpiEstim allows to compute Bayesian $R_t$ estimates efficiently from a time series of case counts. To account for the delay between infection and reporting of cases however, these estimates must be shifted back in time by the mean delay, or one must first infer the time series of infections

in an intermediate step using deconvolution [7, 25]. This and other limitations have motivated the recent development of Bayesian methods which model infections and $R_t$ as latent variables. Here we use a semi-mechanistic renewal model similar to existing tools [26, 27] to estimate $R_t$ directly from observed case counts (Fig 1A). This allows us to specify a single generative model for our data, which can then be adapted to right-truncated and incomplete line lists.

Specifically, we model $C_t$, the number of cases by date of report, as Poisson distributed with rate $\lambda_t$, the expected number of cases reported on date $t$. We link $\lambda_t$ to the number of infections on date $t$, denoted $I_t$, via convolution over previous dates, i. e.

$$C_t|\lambda_t \sim \text{Poisson}(\lambda_t), \quad \lambda_t = \sum_{d=0}^{D} I_{t-d}\, \rho_{t-d}\, p_d \tag{1}$$

where $\rho_{t-d}$ is an ascertainment proportion, i. e. the proportion of infections at time $t - d$ that eventually become reported cases, and $p_d$ the probability of report $d$ days after infection, with an assumed maximum delay $D$. We here let $\rho_t = \rho$ for all $t$, i. e. we assume that the ascertainment proportion is constant over time. Importantly, both $\rho$ and $p$ are assumed to be known, and $p$ is typically composed of i) a delay from infection to symptom onset (i. e. the incubation period), and ii) a delay from symptom onset to reporting. The latter is specific to each reporting system and should be estimated from line list data. We here follow the common practice of using the empirical distribution of delays observed in the line list as a proxy for the reporting delay. This naive approach ignores the right truncation of observations and assumes a fixed delay over time, and we assess the bias resulting from this simplification on synthetic data. We remark that a rough estimate of $\rho$ is often sufficient for estimating $R_t$, as constant-in-time multipliers cannot bias $R_t$ estimates and will distort uncertainty quantification only in extreme cases of misspecification [11]. If $\rho_t$ changes over time, however, $R_t$ estimates will be biased, with slow, constant changes introducing small but continuous bias, and incidental abrupt changes introducing bigger but only temporary bias. In real-world settings, both of these types of changes can occur.

The number of infections $I_t$ is modeled via a renewal process as in previous work [2, 28–30], i. e.

$$I_t|\iota_t \sim \text{Poisson}(\iota_t), \quad \iota_t = E[I_t] = R_t \sum_{s=1}^{G} \psi_s\, I_{t-s}, \tag{2}$$

where $\psi$ represents an assumed intrinsic generation time distribution [31] and $G$ is the maximum generation time (see S1 Appendix A.2.2 for the modeling of initial infections). Note that we here use a stochastic formulation, i. e. we account for variance in the offspring distribution by sampling realized infections $I_t$ as independently Poisson distributed given $\iota_t$ [28, 32]. As a smoothing prior for $R_t$ over time, we use a first-order random walk. To ensure $R_t > 0$, we apply a softplus link to the random walk, i. e.

$$\text{softplus}^{-1}(R_t)|R_{t-1} \sim N\big(\text{softplus}^{-1}(R_{t-1}), \sigma_R^2\big), \tag{3}$$

where $\text{softplus}(x) = \frac{\log(1+e^{kx})}{k}$ is the softplus function with sharpness parameter $k$. We choose $k = 4$ to ensure that the random walk is effectively on the unit scale for all realistic values of $R_t$, thereby modeling changes in $R_t$ over time as symmetric (see S1 Appendix A.1.3). This reflects our expectation that interventions and behavioral changes have roughly symmetric effects on $R_t$ [33], but other link functions such as log [26] or generalized logit [27] are also possible. We use weakly informative priors for $R_1$ and $\sigma_R$ (see S1 Appendix A.1.1), to allow the intercept and variance of the random walk to be estimated from the data.

Using Eqs (1)–(3) (full model in S1 Appendix B.1), we can compute the likelihood $P(C_{\{t|t\leq T\}} \mid R_{\{t|t\leq T\}}, \theta)$ with nuisance parameters $\theta$ and, together with corresponding priors (S1 Appendix B.6), sample from the posterior $P(R_{\{t|t\leq T\}}, \theta \mid C_{\{t|t\leq T\}})$ using Markov chain Monte Carlo (MCMC, see Section 2.5) to obtain Bayesian estimates of $R_t$ over time.

## 2.2 (B) Truncation adjustment

The direct estimation of $R_t$ from case counts by date of report $C_t$ requires a deconvolution of $C_t$ with potentially long and difficult-to-estimate delays between infection and reporting. An alternative is to estimate $R_t$ from the total case count by date of symptom onset over all delays, $N_t = \sum_{d=0}^{D} N_{t,d}$ (Fig 1B), such that the delay distribution in Eq (1) is simply the incubation period, which is shorter and generally independent of the reporting system. On the present date $T$ however, only $N_{t,d}$ with $t + d \leq T$ are known, such that the real-time count $\sum_{d=0}^{T-t} N_{t,d}$ is a downward-biased estimate of $N_t$. This bias must be corrected by estimating the number of symptom onsets which have occurred but are not yet reported, also known in the statistical literature as "nowcasting" in the narrow sense [13]. Conceptually this task is related to time-to-event analysis [34], as it requires estimating the distribution of reporting delays from right truncated observations $N_{t,d}$ to predict the not-yet-reported cases.

**Stepwise approach.** To nowcast $R_t$, existing methods follow a stepwise approach [17–19], where first $N_t$ is estimated using the observed $N_{t,d}$ in a truncation adjustment step, and then $R_t$ is estimated from the time series of $N_t$ (instead of $C_t$, as in Section 2.1). For the truncation adjustment step, we specify a model similar to Günther et al. [17], which jointly combines Bayesian smoothing of the epidemic curve [15, 35] with a discrete time-to-event model for the reporting delay distribution [14]. The latter allows to estimate reporting delays that change over time (e. g. due to changes in the reporting process or the burden on health authorities) and are subject to seasonality (e. g. due to weekday effects), which is both observed in practice [14]. For this, we define $p_{t,d}$ as the probability that a case with symptom onset on date $t$ is reported with a delay of $d \in [0, 1, \ldots D]$ days after symptom onset, i. e. we assume that reporting cannot occur before symptom onset (no negative delays) and is at most $D$ days after symptom onset. It is important to assume a long enough maximum delay $D$ that almost fully covers the true delay distribution and only excludes potential outliers. In practice, this choice can be informed by already observed delays (e. g. the 95% empirical quantile of delays), but should ultimately be based on domain knowledge about the reporting system, especially in the early phase of an outbreak. Cases with delays longer than $D$ should then be completely dropped from the analysis, as setting their delays to $D$ instead can lead to right truncation bias. Given the above definition, we assume that cases with symptom onset on date $t$ are Poisson distributed with rate $\lambda_t$, and their reporting delays are categorically distributed with event probabilities $p_{t,d}$. We can then model the observed numbers of cases with symptom onset on date $t$ and reporting delay $d$ as

$$N_{t,d}|\lambda_t, p_{t,d} \sim \text{Poisson}(\lambda_t \, p_{t,d}), \tag{4}$$

To model the delay probabilities $p_{t,d}$, we use a discrete time-to-event model in which we parameterize the hazard of reporting $h_{t,d}$ via a logistic regression [14, 36], i. e.

$$p_{t,d} = h_{t,d} \prod_{i=1}^{d} (1 - h_{t,d-i}), \quad \text{logit}(h_{t,d}) = \gamma_d + z_t^\top \beta + w_{t+d}^\top \eta, \tag{5}$$

where $\gamma_d$ is the logit baseline hazard for delay $d$, and $z_t$ and $w_{t+d}$ are vectors of covariates with coefficient vectors $\beta$ and $\eta$, respectively. This implements a non-parametric, proportional

hazards model [37] for the reporting delay as in Günther et al., however, we here model both effects by date of symptom onset $z_t^\top \beta$ and effects by date of report $w_{t+d}^\top \eta$. We design $z_t$ to account for changes in the reporting delay of cases over time via a piecewise linear weekly change point model, and $w_{t+d}$ to account for differences in reporting between weekdays (see S1 Appendix A.3.1 for details). As a smoothing prior for $\lambda_t$ over time, we use a first-order random walk on the log scale, i. e.

$$\log(\lambda_t)|\lambda_{t-1} \sim N(\log(\lambda_{t-1}), \sigma_{\log(\lambda)}^2). \tag{6}$$

This is a minimally informed prior assuming a stationary time series of symptom onsets which has often been used in earlier work [15, 17, 35, 38], and we test non-stationary exponential smoothing as an alternative prior in a supplementary analysis (S1 Appendix A.1.2). Analogous to Section 2.1, we use weakly informative priors for $\log(\lambda_1)$ and $\sigma_{\log(\lambda)}$ (see S1 Appendix A.1.1), to estimate the random walk intercept and variance from the data.

Given Eqs (4)–(6) (full model in S1 Appendix B.2), we can compute the likelihood $P\left(N_{\{t,d|t+d\leq T\}}|\lambda_{\{t|t\leq T\}}, p_{\{t,d|t\leq T, d\leq D\}}, \theta\right)$, and obtain posterior samples $\hat{\lambda}_{\{t|t\leq T\}}^{(i)}, \hat{p}_{\{t,d|t\leq T, d\leq D\}}^{(i)} \sim P\left(\lambda_{\{t|t\leq T\}}, p_{\{t,d|t\leq T, d\leq D\}}, \theta \mid N_{\{t,d|t+d\leq T\}}\right)$ via MCMC. These can then be used to produce samples of the not-yet-observed case counts $\hat{N}_{\{t,d|t+d>T\}}^{(i)}$ by drawing from Poisson$(\hat{\lambda}_t^{(i)} \hat{p}_{t,d}^{(i)})$, respectively. Samples from the nowcast for $N_t$ are then simply obtained by summing up the already observed and the sampled, not-yet-observed case counts, i. e.
$\hat{N}_t^{(i)} = \sum_{d=0}^{\min(T-t,D)} N_{t,d} + \sum_{d=T-t+1}^{D} \hat{N}_{t,d}^{(i)}.$

To obtain a nowcast for $R_t$, existing nowcasting approaches typically estimate $R_t$ from the nowcast for $N_t$ in a separate step. Since the nowcast for $N_t$ is subject to considerable uncertainty on dates close to the present, stepwise approaches must repeat the $R_t$ estimation step on many samples $\hat{N}_{\{t|t\leq T\}}^{(i)}$ from the truncation adjustment, and finally combine the resulting $R_t$ estimates. This "resampling" procedure is necessary to account for the uncertainty from the truncation adjustment but introduces computational overhead especially if the individual $R_t$ estimation steps are costly. In practice, stepwise approaches therefore either ignore uncertainty from the truncation adjustment [39] or use simple, non-parametric methods such as EpiEstim to estimate $R_t$ [17, 18, 25]. To ensure a consistent comparison of approaches in this study, we fitted the generative model introduced in Section 2.1 on 50 different samples $\hat{N}_{\{t|t\leq T\}}^{(i)}$ using MCMC, which incurred a relevant proportion of the overall computational cost of the stepwise approach (see Section 3.1.6). We additionally produced nowcasts using EpiEstim for $R_t$ estimation in a supplementary analysis (S1 Appendix A.2.4 and C.1).

**Generative approach.** As an alternative to the stepwise approach, we here propose to integrate $R_t$ estimation and truncation adjustment in a single generative model. This is achieved by replacing Eq (6) in the truncation adjustment model with

$$\lambda_t = \sum_{s=0}^{L} I_{t-s} \rho_{t-s} \tau_s, \tag{7}$$

where $\rho_{t-s}$ is again an assumed ascertainment proportion, $\tau$ is the incubation period distribution, $L$ is the maximum incubation period, and $I_t$ is the number of infections on date $t$. Analogous to Section 2.1, infections $I_t$ are then modeled via a renewal model using Eqs (2) and (3). By doing so, we replace the non-parametric smoothing prior for $\lambda_t$ with a model of symptom onsets that arise from latent infections generated through a renewal process. This yields a single hierarchical model based on Eqs (2)–(5) and (7) (full model in S1 Appendix B.3), under

which we can directly compute the joint likelihood

$$P\Big(N_{\{t,d|t+d\leq T\}} \mid \lambda_{\{t|t\leq T\}}, p_{\{t,d|t\leq T, d\leq D\}}, R_{\{t|t\leq T\}}, \theta\Big),$$

and obtain posterior samples for $R_t$ and $N_t$ as described before. The resulting generative approach allows to nowcast $R_t$ and $N_t$ in a single step, while quantifying uncertainty both from truncation adjustment and $R_t$ estimation. Moreover, by explicitly linking the time series of symptom onsets to the underlying infection dynamics, we use the renewal model as a structural prior for the temporal correlation of $N_t$. This way, epidemic modeling is not only used to estimate $R_t$ but also to regularize the nowcast for $N_t$, thereby avoiding the additional non-parametric smoothing of $\lambda_t$ in the stepwise approach.

## 2.3 (C) Missing date imputation

In practice, line list data are often incomplete, i. e. the date of symptom onset can be missing for a substantial proportion of cases. Ignoring incomplete cases and computing nowcasts as described in Section 2.2 only using the complete cases $N_{t,d}^{\text{known}}$ will underestimate $N_t$ and bias nowcasts of $R_t$ if the proportion of cases with missing onset date changes over time. It is therefore important to also include the incomplete cases, which are only available as case counts $C_t^{\text{missing}}$ by date of report. Here we discuss two stepwise approaches of imputing missing onset dates and then show how a missingness model can be directly integrated into the generative nowcasting model (Fig 1C).

**Stepwise approach.**   A simple but potentially biased approach is to treat the reporting delays of cases in the line list as independent and identically distributed. Under this assumption, missing onset dates can be imputed by simply subtracting a random delay from the date of report of each incomplete case, i. e. each case counted in $C_t^{\text{missing}}$. The random delays could for example be drawn from the empirical delay distribution of delays from complete cases in the line list. However, even if there was no right truncation or changes in the reporting delay over time, such a procedure will yield biased results during an epidemic wave. This is because subtracting delays from the date of report implicitly conditions the delay distribution on the date of report [7, 40]. The resulting distribution, also known as backward delay distribution, depends on the shape of the past epidemic curve and therefore varies during an epidemic. A more accurate approach is therefore to only assume that cases with the same date of report have identically distributed delays, and to estimate the so-called backward delay probability $p_{t,d}^{\leftarrow}$ that a case reported on date $t$ had its symptom onset $d$ days ago [17–19]. We here implement this approach by modeling the number of cases with known symptom onset date as

$$(N_{t,0}^{\text{known}}, \ldots, N_{t-D,D}^{\text{known}}) \sim \text{Multinom}(p_{t,0}^{\leftarrow}, \ldots, p_{t,D}^{\leftarrow}), \qquad (8)$$

and parameterizing $p_{t,d}^{\leftarrow}$ using an identical discrete time-to-event model as specified in Eq (5), but indexed by date of report instead of date of symptom onset (S1 Appendix B.4). We remark that by conditioning on the date of report, the above model is not biased by right truncation, but potentially by left truncation (see S1 Appendix A.3.2). We compute the likelihood $P(N_{\{t,d|t+d\leq T\}}^{\text{known}} \mid p_{t,d}^{\leftarrow}, \theta)$ to obtain posterior samples from $P(p_{t,d}^{\leftarrow}, \theta \mid N_{\{t,d|t+d\leq T\}}^{\text{known}})$ via MCMC, and use the latter to impute missing onset dates by drawing random delays from the corresponding backward delay distribution. Importantly, using observed delays to impute missing onset dates implies a missing-at-random assumption, i. e. reporting delays are assumed to be independent of whether cases in the line list are complete or incomplete. Moreover, both the independent imputation and the backward-delay imputation approach are stepwise in that they add an additional imputation step prior to nowcasting. As a result, the imputed symptom onset dates

are treated as observed data during the following step(s), which creates the same obstacle to uncertainty quantification as described in Section 2.2. Thus, to account for uncertainty of the imputation, a multiple imputation scheme must be used, where multiple imputed data sets are created and a separate nowcast is estimated for each data set [18]. With more complex nowcasting models however, multiple imputation is often impractical and therefore omitted [17], as fitting to a large number of imputed data sets would be prohibitively computationally expensive. Similarly, we only assessed single imputations due to resource constraints in this study.

**Generative approach.** As an alternative, we propose to directly account for missing symptom onset dates in the generative nowcasting model described in Section 2.2. For this, we assume that symptom onsets on date $t$ become known with probability $\alpha_t$ and missing with probability $1 - \alpha_t$. The case counts with known and missing symptom onset date $t$ and delay $d$ are then modeled as

$$N_{t,d}^{\text{known}}|\lambda_t, p_{t,d}, \alpha_t \sim \text{Poisson}(\lambda_t \, p_{t,d} \, \alpha_t), \quad N_{t,d}^{\text{missing}}|\lambda_t, p_{t,d}, \alpha_t \sim \text{Poisson}(\lambda_t \, p_{t,d} \, (1 - \alpha_t)), \quad (9)$$

which again assumes that symptom onset dates are missing at random, i. e. independent of their reporting delay. While the $N_{t,d}^{\text{missing}}$ are not directly observed in the line list, we know that $C_t^{\text{missing}} = \sum_{d=0}^{\min(D,t-1)} N_{t-d,d}^{\text{missing}}$. Hence, $C_t^{\text{missing}}$ is the sum of Poisson distributed case numbers from previous days with corresponding delays, and it is also Poisson distributed with

$$C_t^{\text{missing}}|\lambda, p, \alpha \sim \text{Poisson}\left(\sum_{d=0}^{\min(D,t-1)} \lambda_{t-d} \, p_{t-d,d} \, (1 - \alpha_{t-d})\right). \quad (10)$$

Note that under this model, only observations $C_t^{\text{missing}}$ for $t > D$ should be used to avoid bias from left-truncation, or the latent parameters $\lambda$, $p$, and $\alpha$ must be further extended into the past (S1 Appendix A.4). As a minimally informed, stationary smoothing prior for $\alpha_t$ over time, we use a first-order random walk on the logit scale, i. e.

$$\text{logit}(\alpha_t)|\alpha_{t-1} \sim N(\text{logit}(\alpha_{t-1}), \sigma_{\text{logit}(\alpha)}^2), \quad (11)$$

and estimate $\alpha_1$ and $\sigma_{\text{logit}(\alpha)}$ with weakly informed priors as before. This assumes that the probability of missing symptom onset dates varies only by date of symptom onset, but more sophisticated models could also account for weekday effects or effects by date of report. Assuming $N_{t,d}^{\text{known}}$ and $C_t^{\text{missing}}$ as independent given $\lambda$, $p$, and $\alpha$, we can replace Eq (4) in the existing hierarchical nowcasting model described in Section 2.2 with Eqs (9), (10) and (11) (full model in S1 Appendix B.5). Under the resulting model, we can compute the joint likelihood

$$P\left(N_{\{t,d|t+d\leq T\}}^{\text{known}}, C_{\{t|t\leq T\}}^{\text{missing}} \mid \lambda_{\{t|t\leq T\}}, p_{\{t,d|t\leq T, d\leq D\}}, \alpha_{\{t|t\leq T\}}, R_{\{t|t\leq T\}}, \theta\right)$$

and obtain posterior samples for $R_t$ and $N_t$ using MCMC as before. Therefore, the model can be jointly fit to incomplete line list data without the need for an additional imputation step. This also accounts for the uncertainty resulting from missing onset dates without the use of a multiple imputation scheme. In comparison to stepwise approaches, the generative approach requires estimating the probability of missing onset dates over time, but in turn avoids estimating backward-delay distributions. Moreover, using estimates for $\alpha_t$, separate nowcasts for $N_t^{\text{known}}$ and $N_t^{\text{missing}}$ can be easily obtained.

## 2.4 Modeling of overdispersion

As found in earlier work [17], the modeling of overdispersed case counts can improve now-casting performance in real-world settings. For the models presented in this study, overdispersion can be accounted for by modeling case counts as Negative Binomial instead of Poisson distributed, i. e. by using

$$N_{t,d}|\lambda_t, p_{t,d}, \phi \sim \text{NegBin}(\lambda_t\, p_{t,d}, \phi), \tag{12}$$

instead of Eq 4, where the inverse of $\phi$ defines the level of overdispersion pooled for all observations, with $Var[N_{t,d}|\lambda_t, p_{t,d}, \phi] = (\lambda_t\, p_{t,d})\left(1 + \frac{\lambda_t\, p_{t,d}}{\phi}\right)$. We use Eq 12 when nowcasting COVID-19 hospitalizations in Switzerland, and place a weakly informative prior on $\frac{1}{\sqrt{\phi}}$ to regularize the overdispersion parameter (S1 Appendix E.3).

## 2.5 Implementation and estimation

We implemented all models used in the different nowcasting approaches (Fig 1) as probabilistic programs in Stan [41] (S1 Appendix C.1). This allowed us to estimate all parameters in a fully Bayesian framework with Markov chain Monte Carlo (MCMC) using cmdstan version 2.30.0 via cmdstanr version 0.5.0 [42]. To provide regularization, we used weakly informative priors on the model parameters (see S1 Appendix B.6 for details). For each estimation, the default configuration of the No-U-Turn sampler was run, i. e. four chains with 1,000 warm-up and 1,000 sampling iterations each. After model fitting, we computed common Bayesian model diagnostics to check for convergence (Gelman-Rubin convergence diagnostic $\hat{R}$) [43] and sufficient sample sizes (ESS ratio) [44] (S1 Appendix C.3). We used the R package EpiEstim [8] for $R_t$ estimation with the method by Cori et al., and the R package scoringutils [45] for computation of weighted interval scores. All processing steps and analyses were performed in R 4.1.0, and the models and code for this study are available from zenodo at https://doi.org/10.5281/zenodo.8279675.

## 2.6 Comparison of approaches using synthetic data

We used synthetic data to compare the different stepwise and generative nowcasting approaches. To simulate synthetic data, we generated infections through a stochastic renewal process over a period of 200 days in which $R_t$ follows a piecewise linear time series with pre-specified change points. The simulation matches the generative models described in Section 2.1 and 2.2, but simulates the reporting of individual cases and therefore does not assume independent observation noise. We simulated two different scenarios, representing a first and second epidemic wave, respectively (S1 Appendix D.1.1). The first wave scenario starts with a small number of seeding infections and strong transmission ($R_t = 2$), followed by a shift to suppression ($R_t = 0.8$). The second wave scenario starts with moderate case numbers and controlled transmission ($R_t = 1$), followed by a resurgence ($R_t = 1.4$) and finally decline ($R_t = 0.8$). For each scenario, we produced 50 simulation runs at daily resolution with different realizations of the infection process. We parameterized the generation interval and the incubation period distributions based on estimates for COVID-19 from the literature [46, 47], and supplied the same distributions to the nowcasting models. To mimic the ascertainment of hospitalization data, we let infections become reported hospitalizations with probability $\rho = 2\%$. This is in line with case-hospitalization proportions observed in Switzerland after accounting for an under-ascertainment factor of 2.3–3.1 infections per confirmed case, as estimated by a Swiss seroprevalence study [48]. For each simulation run, we obtained a synthetic line list by

simulating the symptom onset and day of report of each case. The baseline reporting delay was specified as a discretized gamma distribution with a mean of 9 days and standard deviation of 8 days, which is in line with the delay between symptom onset and reporting of hospitalization during the first year of the COVID-19 pandemic in Switzerland (Fig S in S1 Appendix). Under such a reporting delay, approximately 50% of hospitalizations are reported within one week after symptom onset, and 80% after two weeks. We used a piecewise linear model to simulate changes in the reporting hazard over time, with random changes in the trend every four weeks, which differed across simulation runs. We also included weekday effects with the odds of reporting being at the baseline Wednesday—Friday, 70% lower on Saturdays, 80% lower on Sundays, 10% higher on Mondays, and 5% higher on Tuesdays. Details of the simulation of infections and reporting are provided in S1 Appendix D.

For each simulation run and scenario, nowcasts were obtained for selected weeks in different phases of the epidemic wave (Figs 2–5, S1 Appendix B.6) by applying the nowcasting approaches at different lags of the respective week (at the end of the week, one week after, two weeks after). This allowed us to assess changes in performance as more data becomes available. The nowcasts had a daily resolution and we summarized their performance at a weekly level. We fitted the models using always the last three months of line list data until the date of the nowcast. Complete line list data, i. e. without missing symptom onset dates, were used to compare the direct $R_t$ estimation with the stepwise and generative truncation adjustment approaches. Incomplete line list data were used to compare the stepwise and the generative missing date imputation approaches. Here we randomly selected cases to have a missing symptom onset date with a probability varying over time between 20% and 60% according to a random piecewise linear function, which differed across simulation runs.

To evaluate nowcasting performance in each of the 50 simulation runs, we scored the nowcasts for each week in each phase with respect to $N_t$ and $R_t$ against the simulated ground truth. As a performance measure, we used the weighted interval score (WIS) [49], which is a quantile-based proper scoring rule that approximates the continuous ranked probability score (CRPS) [50]. The CRPS generalizes the absolute error to probabilistic nowcasts. For an observation $y$, i. e. the true $N_t$ or $R_t$, and a predictive distribution $F$, i. e. the probabilistic nowcast for $N_t$ or $R_t$, the weighted interval score is computed as

$$\text{WIS}(F, y) = \frac{1}{2K+1} \left( |y - F_{0.5}| + \sum_{k=1}^{K} (\alpha_k \, \text{IS}_{\alpha_k}(F, y)) \right), \tag{13}$$

where $|y - F_{0.5}|$ is the absolute error of the median, $\alpha_k \in \{\alpha_1, \ldots, \alpha_K\}$ represent different central prediction intervals at coverage level $(1 - \alpha_k)$, and $\text{IS}_\alpha(F, y)$ is the interval score for a specific interval, defined as

$$\text{IS}_\alpha(F, y) = \underbrace{\left( F_{1-\frac{\alpha}{2}} - F_{\frac{\alpha}{2}} \right)}_{dispersion} + \underbrace{\frac{2}{\alpha} \left( y - F_{1-\frac{\alpha}{2}} \right) \mathbb{1}_{y > F_{1-\frac{\alpha}{2}}}}_{underprediction} + \underbrace{\frac{2}{\alpha} \left( F_{\frac{\alpha}{2}} - y \right) \mathbb{1}_{y < F_{\frac{\alpha}{2}}}}_{overprediction}. \tag{14}$$

That is, the interval score can be decomposed into three penalty components, where the dispersion component encourages sharpness, and the under- and overprediction components encourage calibration of the nowcast. We used $K = 11$ different $\alpha_k$, corresponding to the 10%, 20%, ..., 90%, 95%, and 98% credible intervals (CrI) for $N_t$ and $R_t$, respectively. To obtain an overall score $\overline{\text{WIS}}$ for each evaluated week, we took the arithmetic mean of scores from each day of the week, which again yields a proper score. We furthermore computed the relative contribution of each penalty component (dispersion, over-, and underprediction) to the respective

$\overline{\text{WIS}}$. In the results, we report the average $\overline{\text{WIS}}$ over all 50 simulation runs, as well as the percentage of runs where each approach performed best.

## 2.7 Application to COVID-19 in Switzerland

We also compared the direct, stepwise, and generative nowcasting approaches during the COVID-19 pandemic in Switzerland. For this, we used anonymized line list data of cases reported to the Federal Office of Public Health (FOPH) from January 01, 2020—March 31, 2021. Because in Switzerland the date of symptom onset for COVID-19 was only consistently recorded for hospitalized patients, we only used cases of hospitalization from the line list. As date of report, the date when a hospitalization was recorded by FOPH was used. The symptom onset date was either reported together with the clinical record of hospitalization or slightly before (due to a previous test result which was later merged), but not updated afterward. Negative delays were therefore assumed to be data entry errors, and the respective symptom onset dates were set to missing.

As the date of symptom onset was missing for a considerable fraction of hospitalizations, and $R_t$ estimates based on hospitalization records can be biased particularly in the beginning of a pandemic and across different populations [6], we were not able to assess the different nowcasting approaches against a ground truth for $N_t$ or $R_t$. We therefore focused on an evaluation using consolidated nowcasts for $N_t$ and a qualitative comparison for $R_t$. We produced retrospective nowcasts for the number of hospitalizations by date of symptom onset $N_t$ (including cases with and without known symptom onset) and the effective reproduction number $R_t$ during different phases of the first and second wave (before, at, and after peak, respectively). We assumed a maximum delay of 8 weeks, which covered 97.03% of reporting delays, and removed the small percentage of cases with larger delays from the analysis. When modeling weekday effects on the reporting delay, we coded national public holidays in Switzerland as Sundays. To account for the gradual replacement of the SARS-CoV-2 wild-type by the alpha variant from December 2020—April 2021, we used different incubation period distributions ($\tau^{\text{wild-type}} = \Gamma(\alpha = 2.74, \beta = 0.52)$ [46], $\tau^{\text{alpha}} = \Gamma(\alpha = 3.08, \beta = 0.63)$ [51]) and generation interval distributions ($\psi^{\text{wild-type}} = \Gamma(\alpha = 1.43, \beta = 0.29)$ [47], $\psi^{\text{alpha}} = \Gamma(\alpha = 1.75, \beta = 0.38)$ [52]) over time according to a logistic growth transition (S1 Appendix E.2).

To assess the performance of nowcasts for the number of symptom onsets, we computed "consolidated" estimates of $N_t$ as a proxy ground truth. That is, while $N_t^{\text{known}}$ can be reliably observed in retrospect after a sufficiently long delay, the number of cases with missing symptom onset date $N_t^{\text{missing}}$ cannot be observed. We therefore obtained estimates for $N_t^{\text{missing}}$ from nowcasts at large lags beyond the respective date (7–14 days more than the maximum delay). These nowcasts are informed by fully reported case counts and thus no longer subject to right truncation. The resulting "consolidated" estimates of $N_t$ were then compared against the different real-time nowcasts, using the same weighted interval scoring rule as described for the synthetic data. Since the effective reproduction number $R_t$ cannot be observed in a real-world setting, and corresponding estimates are subject to considerable uncertainty, we refrained from assessing the performance of $R_t$ nowcasts and focused on a qualitative comparison instead.

## 3 Results

In the following, we present results from an evaluation of the different nowcasting approaches for complete and incomplete line list data. First, the direct, stepwise and generative approaches for nowcasting $R_t$ from truncated line lists are compared on synthetic data based on a simulation of a first and second wave scenario (Section 3.1). Then, two different stepwise approaches

**Table 1. Overview of main results for different nowcasting approaches for complete and incomplete line list data.** Shown is a summary, based on an evaluation on synthetic and real-world data, of the qualitative behaviour of i) different approaches for $R_t$ estimation and truncation adjustment, and ii) different approaches for missing data imputation.

| Task | Approach | Method | Real-time $N_t$ nowcasts | Real-time $R_t$ nowcasts |
|---|---|---|---|---|
| Nowcasting from truncated data (complete line list) | Direct | Simple $R_t$ estimation (no truncation adjustment) | No nowcasts of $N_t$ produced. | Strong downward bias if not adjusted for right truncation. Can produce spurious signals of reduced transmission in all phases of epidemic. |
| | Stepwise | 1) Truncation adjustment 2) $R_t$ estimation | Underprediction during growth before peak & overprediction during decline after peak (if stationary smoothing prior for $\lambda_t$ is used, e.g. random walk). | Bias towards $R_t = 1$ (if stationary smoothing prior for $\lambda_t$ is used, e.g. random walk). Can produce spurious signals of reduced transmission in early phase of epidemic. |
| | Generative | Joint nowcasting model for truncation adjustment + $R_t$ estimation | Fits exponential growth or decline trend before and after peak. Tendency to overpredict at peak. | $R_t$ tends to remain at current level. Changes in transmission only picked up after some delay. |
| Imputation of missing data (incomplete line list) | Stepwise | 1) Independent imputation 2) Nowcasting | Underprediction around peak (incidence curve too flat). Overconfident nowcasts (unless multiple imputation is used, computationally costly). | Potential bias at change points of $R_t$ (transitions too smooth). Ignored uncertainty from missing data (but small compared to overall uncertainty). |
| | Stepwise | 1) Backward imputation 2) Nowcasting | Overconfident nowcasts (unless multiple imputation is used, computationally costly). | Ignored uncertainty from missing data (but small compared to overall uncertainty). |
| | Generative | Joint nowcasting model for missingness + truncation adjustment + $R_t$ estimation | Full representation of uncertainty from missing data. Risk of underprediction if backfilling is present but not modeled. | Uncertainty from missing data accounted for. |

and the generative approach of accounting for missing symptom onset dates in incomplete line lists are compared on synthetic data (Section 3.2). Finally, we present results from an evaluation of the different approaches on real-world data from the COVID-19 pandemic in Switzerland (Section 3.3). Table 1 provides an overview of our main findings.

## 3.1 Synthetic data: $R_t$ estimation and truncation adjustment

We used complete synthetic line list data, i. e. without missing symptom onset dates, to compare i) the direct $R_t$ estimation approach, ii) the stepwise nowcasting approach with an additional truncation adjustment step, and iii) the generative nowcasting approach with a joint truncation and renewal model.

**3.1.1 Nowcasts of the number of cases by symptom onset date $N_t$.** Fig 2 shows the simulated true $N_t$ and corresponding nowcasts for an exemplary simulation run in different phases of the first wave scenario (corresponding results for the second wave scenario are shown in Fig D in S1 Appendix). The baseline simulated reporting delay had a mean of 9 days and standard deviation of 8 days, and the maximum assumed delay was $D = 56$ days. The nowcasts were conducted using the different approaches and at different lags (nowcast at end of same week, one week after, two weeks after). The direct $R_t$ estimation approach did not produce nowcasts for $N_t$, as it directly uses counts by date of report $C_t$ instead.

Compared to $C_t$, which includes reporting delays, the time series of $N_t$ tracked transmission dynamics in a more timely and regular manner (Figs B–C in S1 Appendix). In a real-time setting however, the reported number of symptom onsets can have a substantial downward bias towards the present (Fig 2, grey bars), because a proportion of symptom onsets that occur close to the present are not yet reported at the time of nowcasting. This truncation bias was adjusted for both by the stepwise and generative approach, however, nowcasts differed between the two approaches especially on dates close to the present.

While the generative approach predicted exponentially growing case counts before the peak (Fig 2A.1) and exponentially declining case counts after the peak of the wave (Fig 2A.3), the

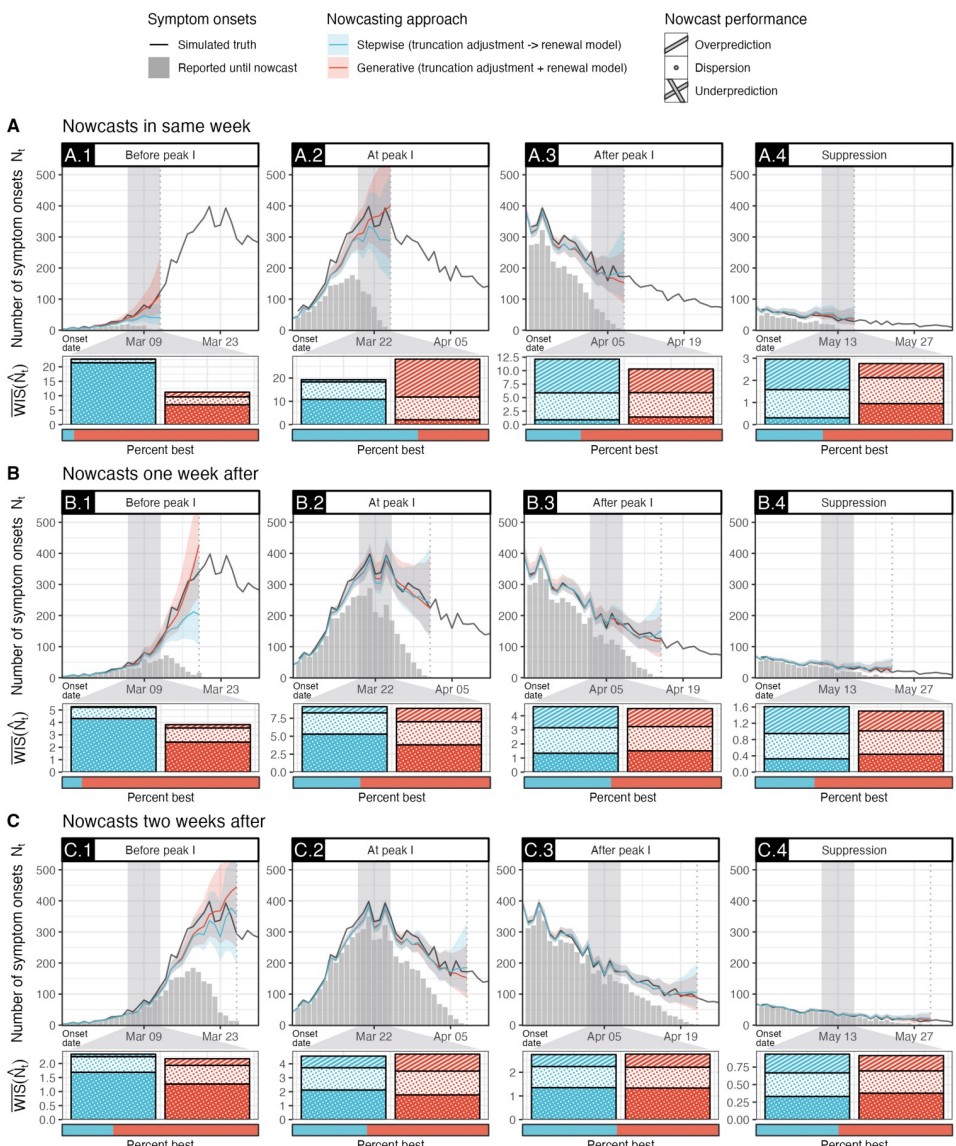

**Fig 2. Nowcasts of $N_t$ on line list data of a simulated first wave scenario using different approaches of adjusting for right truncation.** Shown are the true number of cases by symptom onset date $N_t$ (black), the number of cases reported until the nowcast date (grey bars), and point nowcasts with 95% credible intervals (CrI) in four different phases of the epidemic wave, obtained through i) a stepwise approach using cases by date of symptom onset with a truncation adjustment step (blue), and ii) a generative approach using cases by date of symptom onset with an integrated truncation and renewal model (red). The direct approach using cases by date of report cannot produce nowcast of $N_t$. Shown below each phase is the weighted interval score (WIS, lower is better) for $N_t$ nowcasts of each approach during a selected week (grey shade) over 50 scenario runs (see Table 2 for exact figures). Colored vertical bars show average scores, decomposed into penalties for underprediction (crosshatch), dispersion (circles), and overprediction (stripes). The horizontal bar below shows the percentage of times each approach achieved the lowest WIS out of 50 scenario runs, respectively. Results are shown for nowcasts made at different lags from the selected week (vertical dotted lines), i. e. at the end of the selected week (top row), one week later (middle row), and two weeks later (bottom row).

**Table 2. Performance of nowcasts of $N_t$ on line list data of a simulated first wave scenario using different approaches of adjusting for right truncation.** Shown is the performance of $N_t$ nowcasts in four different phases of the epidemic wave, at different lags of a selected week (same week [0–6 days], one week after [7–13 days], and two weeks after [14–20 days]). Nowcasts were obtained through i) a stepwise approach using cases by date of symptom onset with a truncation adjustment step, and ii) a generative approach using cases by date of symptom onset with an integrated truncation and renewal model. Performance is measured by the weighted interval score (WIS, lower is better), shown is the average score over 50 scenario runs ($\overline{\text{WIS}}$) and the percentage of runs in which each approach achieved the best score (%$^{\text{best}}$).

| Phase | Lag [days] | Stepwise | | Generative | |
|---|---|---|---|---|---|
| | | $\overline{\text{WIS}}(\hat{N}_t)$ | %$^{\text{best}}$ | $\overline{\text{WIS}}(\hat{N}_t)$ | %$^{\text{best}}$ |
| Before peak I | 0–6 | 22.65 | 6% | 11.23 | 94% |
| | 7–13 | 5.27 | 10% | 3.82 | 90% |
| | 14–20 | 2.33 | 26% | 2.17 | 74% |
| At peak I | 0–6 | 19.28 | 64% | 28.10 | 36% |
| | 7–13 | 9.12 | 34% | 8.88 | 66% |
| | 14–20 | 4.55 | 38% | 4.69 | 62% |
| After peak I | 0–6 | 12.13 | 28% | 10.31 | 72% |
| | 7–13 | 4.64 | 44% | 4.51 | 56% |
| | 14–20 | 2.75 | 48% | 2.77 | 52% |
| Suppression | 0–6 | 2.96 | 34% | 2.76 | 66% |
| | 7–13 | 1.61 | 30% | 1.50 | 70% |
| | 14–20 | 0.94 | 42% | 0.91 | 58% |

stepwise approach predicted case counts to remain at the recent level in all phases. This qualitative difference was strongest for nowcasts conducted at short lags (Fig 2A), where nowcasting uncertainty was especially high, and declined with longer lags (Fig 2B–C), as more data became available for the respective week.

Below each lag and phase, Fig 2 and Fig D in S1 Appendix show the weighted interval score (WIS) for each approach across 50 simulation runs (see Table 2 and Table B in S1 Appendix for quantitative result). The generative approach achieved systematically better scores in phases with exponential growth or decline, while the stepwise approach had a bias towards underprediction of $N_t$ during exponential growth (Fig 2A.1) and towards overprediction during decline (Fig 2A.3). This further highlights the difference in smoothing assumptions of the two approaches. While the renewal model component in the generative approach predicted a continuation of recent infection dynamics, the minimally informed random walk prior in the stepwise approach predicted stationary case counts. Notably, at the peak of the wave, this led the generative approach to overpredict case counts, however only at short lags (same week nowcast, Fig 2A.2), when little signal on the change in transmission dynamics was available. Overall, the WIS for both approaches decreased substantially with larger lags, and nowcasts conducted two weeks after the evaluation period show little difference in performance between the stepwise and the generative approach (Fig 2C).

**3.1.2 Nowcasts of the effective reproduction number $R_t$.** Fig 3 shows results for nowcasting $R_t$ in different phases of the first wave scenario (corresponding results for the second wave scenario are shown in Fig E in S1 Appendix). Here we found distinct biases in the direct and the stepwise approach. The direct $R_t$ estimation approach, which used the empirical distribution of delays between symptom onset and reporting in the line list to model observed cases $C_t$, showed a strong downward bias of $R_t$ nowcasts during the first wave (Fig 3A.1–3A.3), where few cases had been observed initially. This reflects the right truncation of line list data, which generally leads to an underestimation of reporting delays and therefore of infections in the recent past. In the second wave scenario, the effect of right truncation was smaller (Fig E in S1 Appendix). However, due to the long delay between infection and reporting in case counts

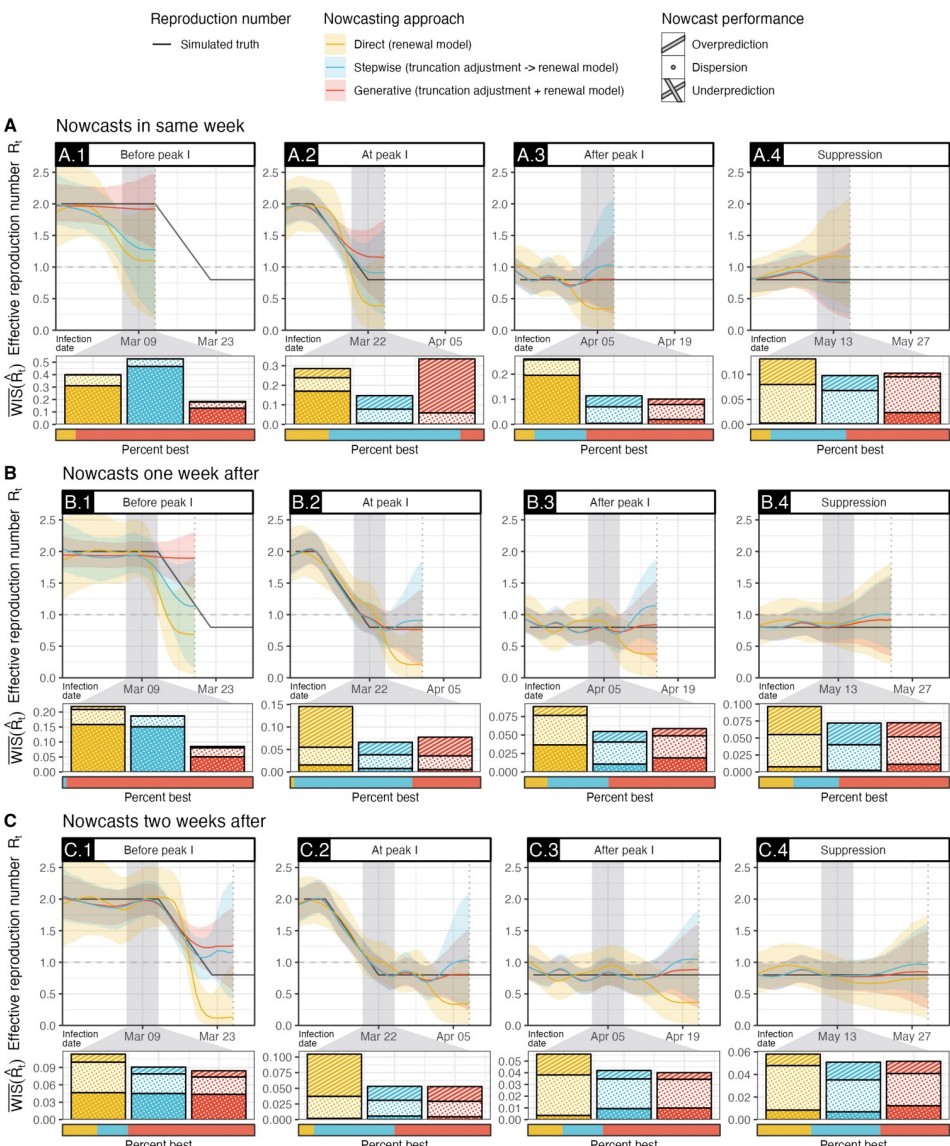

**Fig 3. Nowcasts of $R_t$ on line list data of a simulated first wave scenario using different approaches of adjusting for right truncation.** Shown are the true $R_t$ (black) and point nowcasts with 95% credible intervals (CrI) in four different phases of the epidemic wave, obtained through i) a direct approach using cases by date of report with no truncation adjustment (yellow), ii) a stepwise approach using cases by date of symptom onset with a truncation adjustment step (blue), and iii) a generative approach using cases by date of symptom onset with an integrated truncation and renewal model (red). Shown below each phase is the weighted interval score (WIS, lower is better) for $R_t$ nowcasts of each approach during a selected week (grey shade) over 50 scenario runs (see Table 3 for exact figures). Colored bars show average scores, decomposed into penalties for underprediction (crosshatch), dispersion (circles), and overprediction (stripes). The horizontal bar below shows the percentage of times each approach achieved the lowest WIS out of 50 scenario runs, respectively. Results are shown for nowcasts made at different lags from the selected week (vertical dotted lines), i. e. at the end of the selected week (top row), one week later (middle row), and two weeks later (bottom row).

$C_t$, changes in $R_t$ were detected comparatively late by the direct $R_t$ estimation approach. The stepwise truncation adjustment approach showed a different form of bias, i. e. nowcasts tended to $R_t = 1$ towards the present (Fig 3A.1–3A.4), which translated into under- or overprediction of $R_t$ depending on the epidemic phase. Importantly, this was observed although the $R_t$

**Table 3. Performance of nowcasts of $R_t$ on line list data of a simulated first wave scenario using different approaches of adjusting for right truncation.** Shown is the performance of $R_t$ nowcasts in four different phases of the epidemic wave, at different lags of a selected week (same week [0–6 days], one week after [7–13 days], and two weeks after [14–20 days]). Nowcasts were obtained through i) a direct approach using cases by date of report with no truncation adjustment, ii) a stepwise approach using cases by date of symptom onset with a truncation adjustment step, and iii) a generative approach using cases by date of symptom onset with an integrated truncation and renewal model. Performance is measured by the weighted interval score (WIS, lower is better), shown is the average score over 50 scenario runs ($\overline{\text{WIS}}$) and the percentage of runs in which each approach achieved the best score (%$^{\text{best}}$).

| Phase | Lag [days] | Direct | | Stepwise | | Generative | |
|---|---|---|---|---|---|---|---|
| | | $\overline{\text{WIS}}(R_t)$ | %$^{\text{best}}$ | $\overline{\text{WIS}}(R_t)$ | %$^{\text{best}}$ | $\overline{\text{WIS}}(R_t)$ | %$^{\text{best}}$ |
| Before peak I | 0–6 | 0.40 | 10% | 0.53 | 0% | 0.18 | 90% |
| | 7–13 | 0.22 | 0% | 0.19 | 2% | 0.09 | 98% |
| | 14–20 | 0.11 | 18% | 0.09 | 16% | 0.08 | 66% |
| At peak I | 0–6 | 0.29 | 22% | 0.15 | 66% | 0.34 | 12% |
| | 7–13 | 0.14 | 2% | 0.07 | 62% | 0.08 | 36% |
| | 14–20 | 0.10 | 8% | 0.05 | 42% | 0.05 | 50% |
| After peak I | 0–6 | 0.26 | 10% | 0.11 | 26% | 0.10 | 64% |
| | 7–13 | 0.09 | 12% | 0.06 | 32% | 0.06 | 56% |
| | 14–20 | 0.06 | 18% | 0.04 | 21% | 0.04 | 61% |
| Suppression | 0–6 | 0.13 | 10% | 0.10 | 38% | 0.10 | 52% |
| | 7–13 | 0.10 | 18% | 0.07 | 24% | 0.07 | 58% |
| | 14–20 | 0.06 | 28% | 0.05 | 36% | 0.05 | 36% |

estimation step of the stepwise approach used a smoothing prior which expects $R_t$ to remain at its current level. This indicates that the $R_t$ estimation step was also impacted by the stationary smoothing assumption in the previous truncation adjustment step, i. e. the constant $N_t$ nowcasts biased the downstream $R_t$ estimates towards equilibrium-level transmission. The generative approach did not show such a bias and generally achieved the smallest WIS among the three approaches (Table 3). Where no sufficient signal for a change in transmission dynamics was available, $R_t$ nowcasts from the generative approach remained at the current level. At the peak of the epidemic wave, when transmission declined from $R_t = 2$ to $R_t = 0.8$, this led to an overprediction of $R_t$ for nowcasts in the same week (Fig 3A2). For all approaches, performance scores improved with larger lags, however, the inferior performance of the direct approach in the first wave scenario was pronounced even for nowcasts at a two-week lag (Fig 3C).

**3.1.3 Non-stationary smoothing.** In the above comparisons, a random walk prior on the expected number of symptom onsets was used in the truncation adjustment step of the stepwise approach. While this smoothing prior has been a default choice in existing approaches [15, 17, 35, 38], we also implemented a non-stationary, exponential smoothing prior (S1 Appendix A.1.2) to test whether the observed bias in nowcasts remained. We found that non-stationary smoothing in the stepwise approach reduced the bias of both $N_t$ and $R_t$ nowcasts and improved the WIS, but a slight bias remained during phases of exponential growth or decline (Figs H–K in S1 Appendix). We also tested the use of a non-stationary smoothing prior on $R_t$ for the generative approach, which led to slightly better performance of $R_t$ nowcasts when transmission was changing.

**3.1.4 $R_t$ estimation using EpiEstim.** As it is widely used in practice, we also implemented a stepwise approach where the $R_t$ estimation step is performed using the non-parametric method by Cori et al. via the package EpiEstim (S1 Appendix A.2.4), instead of our semi-mechanistic renewal model described in Sec 2.1. Since the $R_t$ estimation step does not influence the previous truncation adjustment step, nowcasts for $N_t$ with EpiEstim were identical compared to when using a semi-mechanistic renewal model. The nowcasts for $R_t$ showed the

same bias towards $R_t = 1$, however, they were often more volatile and at the same time considerably less uncertain (Figs L–M in S1 Appendix) when using EpiEstim in the $R_t$ estimation step. This overconfidence led to 11% higher on average WIS values of the stepwise approach with EpiEstim (Table J–K in S1 Appendix). We note that when using EpiEstim, nowcasts for $R_t$ could only be obtained at a lag of 8 days or more, as $R_t$ estimates must be shifted backward in time to account for the incubation period and to center the 7-day smoothing window used for EpiEstim [7].

**3.1.5 Misspecification of the incubation period distribution.** All approaches discussed in this study require an assumed incubation period distribution $\tau$, and are susceptible to misspecification of this distribution when estimating $R_t$ [7]. However, the stepwise approach only requires $\tau$ during the $R_t$ estimation step, not during the earlier truncation adjustment step. Thus, in the stepwise approach, nowcasts of $N_t$ cannot be biased by a misspecification of the incubation period distribution. In contrast, the generative approach estimates both $N_t$ and $R_t$ in a joint model, such that nowcasts for $N_t$ can also be influenced by $\tau$. To test whether this makes the generative approach more susceptible to a misspecification of the incubation period distribution, we conducted a sensitivity analysis in which we supplied the nowcasting models with an incubation period distribution that had a 50% shorter (Figs N–O in S1 Appendix) or a 50% longer (Figs P–Q in S1 Appendix) mean than the true distribution. We found neither the stepwise nor the generative approach to have a decreased performance in nowcasting $N_t$ (Figs N, P in S1 Appendix), i. e. nowcasts were highly similar to our main analysis with the correctly specified incubation period distribution. In particular, even with a misspecified $\tau$, the generative approach had less bias than the stepwise approach during the exponential growth or decline phase. Performance with respect to $R_t$ was worse for all approaches (Figs O, Q in S1 Appendix). This can be attributed to a general limitation of $R_t$ estimation methods, i. e. that a misspecified mean incubation period leads to temporal inaccuracy of $R_t$ estimates [7]. Moreover, when the specified incubation period was too short, we observed higher volatility of $R_t$ estimates from all approaches (Fig O in S1 Appendix). Compared to our main analysis, the relative performance of $R_t$ nowcasts between the different approaches remained unchanged.

**3.1.6 Comparison of runtimes.** An important runtime consideration for the stepwise approach is that it requires a resampling scheme where the $R_t$ estimation step is repeatedly applied to posterior samples from the truncation adjustment step to account for the uncertainty of the $N_t$ nowcasts. In our application to the simulated first and second wave scenario (using 4 cores on an AMD EPYC 7763 CPU per method run), a single resampling of the $R_t$ estimation step took on average 1.0 seconds, which is approximately 2.8% of the average runtime of the truncation adjustment step (36.3 seconds). In contrast, the generative approach fits a single model for estimating both $N_t$ and $R_t$, which however had a longer average runtime (241.0 seconds) due to the increased model complexity. The relative performance of both approaches therefore depends on the number of resampling iterations used by the stepwise approach. To match the number of posterior $N_t$ samples of the generative approach, the stepwise approach would need 4000 resampling iterations, however this is likely more than is required for sufficient uncertainty quantification. When using 50 resampling iterations as in our main analysis, the stepwise approach had a total average runtime of 87.3 seconds (with $R_t$ estimation making up 58.2% of the total runtime), thereby being faster than the generative approach (241.0 seconds). However, when using 200 resampling iterations for example, the stepwise approach would have a similar runtime as the generative approach. Here a general difficulty of the stepwise approach is that the number of resampling iterations required to accurately represent the uncertainty of the truncation adjustment step may vary with the scenario and delay distribution, and is difficult to predict in advance. We also note that the

runtime of the MCMC sampling algorithm can generally vary with the complexity of the epidemic trajectory and the amount of noise in the observed case counts.

## 3.2 Synthetic data: Missing symptom onset date imputation

To compare the different stepwise and generative approaches to impute missing onset dates, we used the same synthetic line list data as before, but simulated symptom onset dates to be missing with a time-varying probability $1 - \alpha_t$ (Section 2.6). We tested i) a stepwise approach using independent imputation for the imputation step, ii) a stepwise approach using backward (-delay) imputation for the imputation step, and iii) a generative approach using an integrated missingness model. All three approaches used a generative model for truncation adjustment and $R_t$ estimation as defined in Section 2.2, allowing us to specifically assess differences resulting from the imputation approach.

Fig 4 shows the simulated true $N_t$ and nowcasts for an exemplary simulation run and performance scores over 50 runs in different phases of the first wave scenario (second wave scenario shown in Fig F in S1 Appendix). A strong downward bias in the number of symptom onsets reported until the date of the nowcast was observed both for cases with known and with missing onset date (Fig 4, grey and light grey bars). $N_t$ nowcasts conducted at the end of the same week were highly similar for all three approaches (Fig 4A), however, differences were observed for longer lags, in particular two weeks after (Fig 4C). Here, nowcasts from the generative approach achieved a 26%–78% lower WIS than the stepwise approach (Table 4), mainly due to better representation of the uncertainty from missing onset dates. Nowcasts from the stepwise approach with independent imputation, which ignores the conditioning on the date of report, tended to underpredict the peak of the wave, and this bias increased with longer lags, as more data became available (Fig 4C.2). In contrast, the stepwise approach with backward-delay imputation was generally unbiased, with approximately equal penalties for over- and under-prediction. However, as uncertainty from the imputation step was ignored, this approach yielded overconfident nowcasts at larger lags. Nowcasts from the generative approach generally had the widest uncertainty intervals.

Fig 5 feather miss Rt and Fig G in S1 Appendix show the corresponding results for nowcasts of $R_t$ in the first and second wave scenario, respectively. The three approaches to missing date imputation produced mostly similar $R_t$ nowcasts both at shorter and longer lags, but we observed a small bias of the stepwise approach with independent imputation around the change point of $R_t$ due to the underprediction of the peak in $N_t$ (Fig 5C.2). However, the differences in performance between the approaches were not consistent across phases and comparatively small with regard to the overall WIS (Table 5C.2), which was dominated by the uncertainty and error from the truncation adjustment and $R_t$ estimation.

## 3.3 COVID-19 hospitalizations in Switzerland

We used line list data of hospitalized patients who tested positive for SARS-CoV-2 during the COVID-19 pandemic in Switzerland, reported between January 01, 2020–March 31, 2021, to retrospectively nowcast the number of hospitalizations by date of symptom onset and the effective reproduction number. Overall, the line list contained 25,831 cases of hospitalization, with symptom onset missing for 33% of the cases (Fig R in S1 Appendix). 137 cases had a symptom onset date later than the date of report, in which case the symptom onset date was likely a data entry error and set to missing (see Section 2.7). Our assumed maximum delay of $D = 56$ days covered 97% of observed reporting delays (Fig S in S1 Appendix), and cases with larger delays were removed from the analysis. We assumed different incubation period and generation time distributions for the wildtype and alpha variants of SARS-CoV-2 and

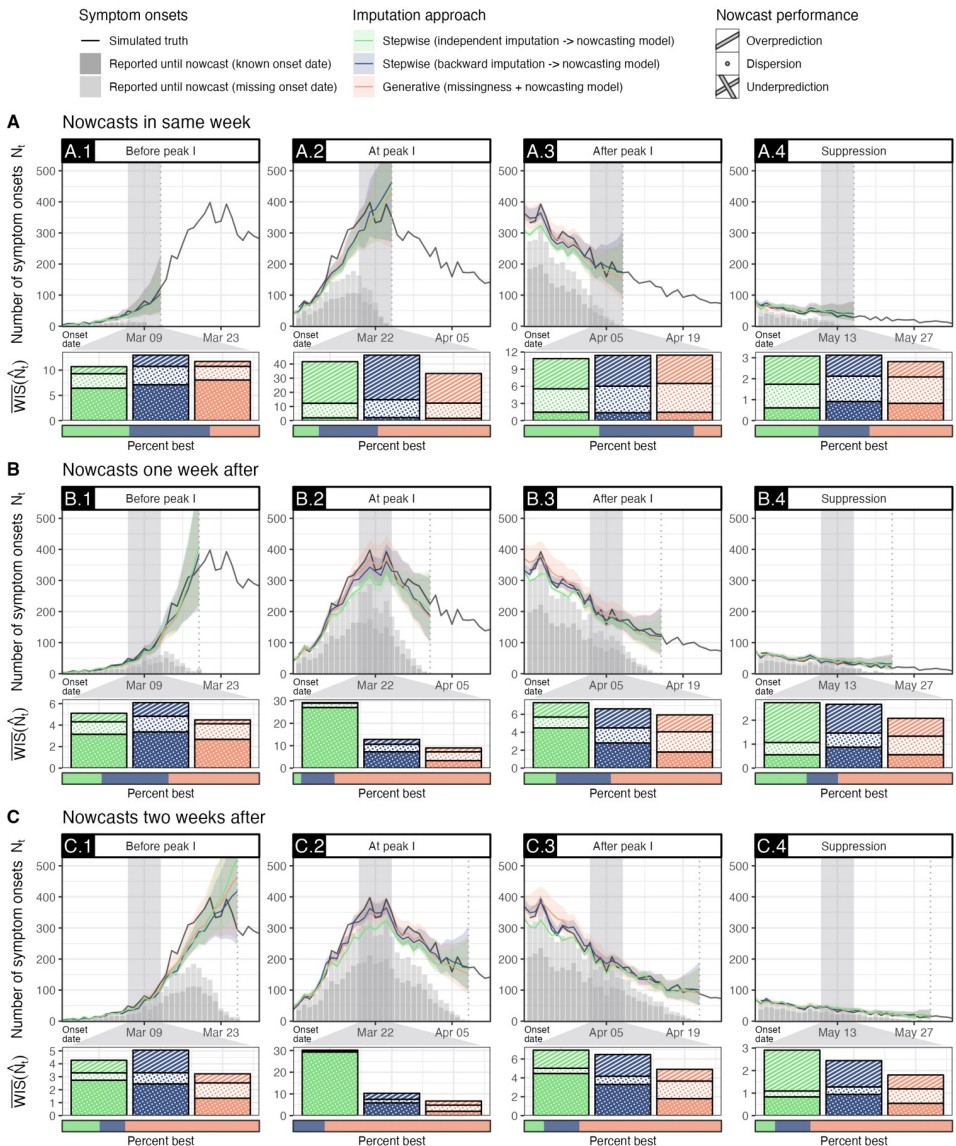

**Fig 4. Nowcasts of $N_t$ on incomplete line list data of a simulated first wave scenario using different approaches to account for missing onset dates.** Shown are the true number of cases by symptom onset date $N_t$ (black), the number of cases reported until the nowcast date (dark grey bars for known, light grey bars for missing onset dates), and point nowcasts with 95% credible intervals (CrI) in four different phases of the epidemic wave, obtained through i) a stepwise approach using an independent imputation step (green), ii) a stepwise approach using a backward imputation step (blue), and iii) a generative approach using an integrated missingness model (red). All approaches used a generative model for nowcasting. Shown below each phase is the weighted interval score (WIS, lower is better) for $N_t$ nowcasts of each approach during a selected week (grey shade) over 50 scenario runs (see Table 4 for exact figures). Colored bars show average scores, decomposed into penalties for underprediction (crosshatch), dispersion (circles), and overprediction (stripes). The horizontal bar below shows the percentage of times each approach achieved the lowest WIS out of 50 scenario runs, respectively. Results are shown for nowcasts made at different lags from the selected week (vertical dotted lines), i. e. at the end of the selected week (top row), one week later (middle row), and two weeks later (bottom row).

**Table 4. Performance of nowcasts of $N_t$ on incomplete line list data of a simulated first wave scenario using different approaches to account for missing onset dates.** Shown is the performance of $N_t$ nowcasts in four different phases of the epidemic wave, at different lags of a selected week (same week [0–6 days], one week after [7–13 days], and two weeks after [14–20 days]). Nowcasts were obtained through i) a stepwise approach using an independent imputation step, ii) a stepwise approach using a backward imputation step, and iii) a generative approach using an integrated missingness model. Performance is measured by the weighted interval score (WIS, lower is better), shown is the average score over 50 scenario runs ($\overline{\text{WIS}}$) and the percentage of runs in which each approach achieved the best score (%$^{\text{best}}$).

| Phase | Lag [days] | Stepwise (independent) | | Stepwise (backward) | | Generative | |
|---|---|---|---|---|---|---|---|
| | | $\overline{\text{WIS}}(\hat{N}_t)$ | %$^{\text{best}}$ | $\overline{\text{WIS}}(\hat{N}_t)$ | %$^{\text{best}}$ | $\overline{\text{WIS}}(\hat{N}_t)$ | %$^{\text{best}}$ |
| Before peak I | 0–6 | 10.69 | 34% | 12.96 | 41% | 11.72 | 25% |
| | 7–13 | 5.10 | 20% | 6.09 | 34% | 4.48 | 46% |
| | 14–20 | 4.27 | 19% | 5.05 | 13% | 3.22 | 68% |
| At peak I | 0–6 | 41.66 | 13% | 46.25 | 30% | 33.31 | 57% |
| | 7–13 | 29.21 | 4% | 12.82 | 17% | 9.01 | 79% |
| | 14–20 | 30.16 | 0% | 10.28 | 16% | 6.71 | 84% |
| After peak I | 0–6 | 10.85 | 38% | 11.42 | 48% | 11.45 | 14% |
| | 7–13 | 7.30 | 16% | 6.61 | 28% | 5.93 | 56% |
| | 14–20 | 6.93 | 10% | 6.47 | 18% | 4.90 | 72% |
| Suppression | 0–6 | 3.08 | 32% | 3.12 | 26% | 2.82 | 42% |
| | 7–13 | 2.73 | 26% | 2.66 | 16% | 2.08 | 58% |
| | 14–20 | 2.91 | 10% | 2.44 | 18% | 1.81 | 72% |

accounted for the introduction and spread of the alpha variant during the second wave in Switzerland (see Section 2.7). Nowcasts were conducted using i) the direct $R_t$ estimation approach, ii) a fully stepwise approach (using a backward-delay imputation step, a truncation adjustment step, and an $R_t$ estimation step), and iii) a fully generative approach (using a joint missingness, truncation, and renewal model).

Fig 6 shows nowcasts of $N_t$, i. e. the number of hospitalizations by date of symptom onset, in different phases of the first and second COVID-19 wave in Switzerland. In contrast to nowcasts by date of hospitalization, these estimates include cases that have developed symptoms but are not yet hospitalized, and thus present a more timely indicator of the underlying infection dynamics. Here we found, similar to our results on synthetic data, that nowcasts from the stepwise approach remained mostly stationary on days close to the nowcast date, while nowcasts from the generative approach extrapolated the recent trend of exponential growth or decline (Fig 6A.1–6A.6). To assess real-time nowcasting performance with regard to $N_t$, we used consolidated estimates based on nowcasts lagged by 7–14 days beyond the maximum delay as a proxy ground truth (see Section 2.7). Based on the consolidated estimates, the generative approach better captured the exponential growth or decline of case numbers and thereby achieved a better WIS in all phases (37–67% lower WIS for generative nowcasts in same week) except at the peak of the second wave (65% lower WIS for stepwise nowcasts in same week). Nowcasts from the stepwise approach were slightly sharper than from the generative approach at larger lags, likely because of the neglected uncertainty from the missing onset date imputation. Of note, both nowcasting approaches underpredicted $N_t$ in the early phase of the epidemic in Switzerland (Fig 6A.1–6C.1), presumably due to an underestimation of the reporting delay based on very few initial hospitalization cases. Fig 7 shows corresponding nowcasts for the hospitalization-based $R_t$. Generally, we found $R_t$ nowcasts at a lag below one week to be largely dominated by the smoothing prior of the nowcasting model (Fig 7A). This is partly to be expected, as transmission dynamics are additionally delayed by the incubation period, which was approximately 5 days on average in our case, thus real-time estimates for $R_t$ are

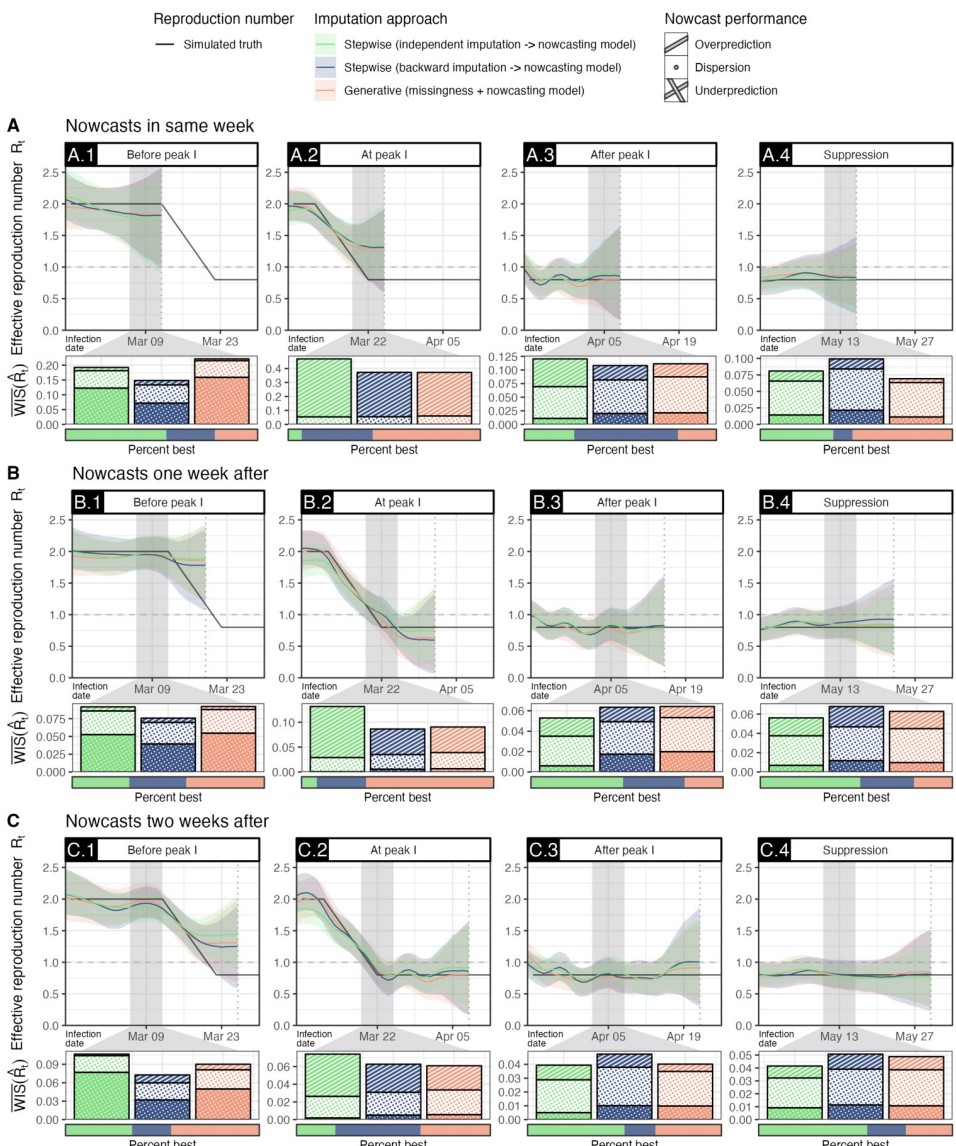

**Fig 5. Nowcasts of $R_t$ on incomplete line list data of a simulated first wave scenario using different approaches of accounting for missing onset dates.** Shown are the true $R_t$ (black) and point nowcasts with 95% credible intervals (CrI) in four different phases of the epidemic wave, obtained through i) a stepwise approach using an independent imputation step (green), ii) a stepwise approach using a backward imputation step (blue), and iii) a generative approach using an integrated missingness model (red). All approaches used a generative model for nowcasting. Shown below each phase is the weighted interval score (WIS, lower is better) for $R_t$ nowcasts of each approach during a selected week (grey shade) over 50 scenario runs (see Table 5 for exact figures). Colored bars show average scores, decomposed into penalties for underprediction (crosshatch), dispersion (circles), and overprediction (stripes). The horizontal bar below shows the percentage of times each approach achieved the lowest WIS out of 50 scenario runs, respectively. Results are shown for nowcasts made at different lags from the selected week (vertical dotted lines), i. e. at the end of the selected week (top row), one week later (middle row), and two weeks later (bottom row).

generally less informed than for $N_t$. Of note, nowcasts using the direct $R_t$ estimation approach, which is based on the more delayed case counts $C_t$, were dominated by the smoothing prior even longer, until a lag of two weeks or more. $R_t$ nowcasts from the fully stepwise and fully generative approach mostly agreed after a lag of two weeks (Fig 7C) but had different trends at

**Table 5. Performance of nowcasts of $R_t$ on incomplete line list data of a simulated first wave scenario using different approaches to account for missing onset dates.** Shown is the performance of $R_t$ nowcasts in four different phases of the epidemic wave, at different lags of a selected week (same week [0–6 days], one week after [7–13 days], and two weeks after [14–20 days]). Nowcasts were obtained through i) a stepwise approach using an independent imputation step, ii) a stepwise approach using a backward imputation step, and iii) a generative approach using an integrated missingness model. Performance is measured by the weighted interval score (WIS, lower is better), shown is the average score over 50 scenario runs ($\overline{\text{WIS}}$) and the percentage of runs in which each approach achieved the best score (%$^{\text{best}}$).

| Phase | Lag [days] | Stepwise (independent) | | Stepwise (backward) | | Generative | |
|---|---|---|---|---|---|---|---|
| | | $\overline{\text{WIS}}(R_t)$ | %$^{\text{best}}$ | $\overline{\text{WIS}}(R_t)$ | %$^{\text{best}}$ | $\overline{\text{WIS}}(R_t)$ | %$^{\text{best}}$ |
| Before peak I | 0–6 | 0.19 | 52% | 0.15 | 25% | 0.22 | 22% |
| | 7–13 | 0.09 | 30% | 0.08 | 30% | 0.09 | 41% |
| | 14–20 | 0.11 | 34% | 0.07 | 20% | 0.09 | 45% |
| At peak I | 0–6 | 0.47 | 7% | 0.37 | 37% | 0.37 | 56% |
| | 7–13 | 0.13 | 8% | 0.09 | 26% | 0.09 | 67% |
| | 14–20 | 0.07 | 20% | 0.06 | 44% | 0.06 | 36% |
| After peak I | 0–6 | 0.12 | 26% | 0.11 | 54% | 0.11 | 20% |
| | 7–13 | 0.05 | 48% | 0.06 | 32% | 0.06 | 20% |
| | 14–20 | 0.04 | 50% | 0.05 | 16% | 0.04 | 34% |
| Suppression | 0–6 | 0.08 | 38% | 0.10 | 10% | 0.07 | 52% |
| | 7–13 | 0.06 | 52% | 0.07 | 28% | 0.06 | 20% |
| | 14–20 | 0.04 | 56% | 0.05 | 20% | 0.05 | 24% |

shorter lags (Fig 7A and 7B), i. e. nowcasts from the stepwise approach tended toward $R_t = 1$, while nowcasts from the generative approach predicted $R_t$ to remain at its current level. This difference in behaviour matched our results on synthetic data. Nowcasting uncertainty was mostly similar between the stepwise and the generative approach.

## 4 Discussion

In this paper, we developed a fully generative model to nowcast the effective reproduction number $R_t$ from right-truncated line list data with missing symptom onset dates. Previous methods have addressed this task by separating missing onset date imputation, truncation adjustment, and $R_t$ estimation into consecutive, independent steps [17–19]. Here we proposed to unify these tasks in a single generative model that can be fit directly to observed data. With such a model, all quantities of interest and their associated uncertainty can be estimated jointly in a Bayesian framework. This is in contrast to stepwise approaches, which face difficulties in propagating uncertainty between consecutive steps and often require resampling schemes that can restrict the individual components. Moreover, information can only flow in one direction, such that priors and model structure from later steps cannot inform earlier steps. Instead, our generative model enables shared regularization between imputation, truncation adjustment, and $R_t$ estimation, thereby eliminating the need for non-parametric smoothing at intermediate steps.

To compare the generative and the stepwise approaches with regard to performance and qualitative behavior, we applied them to synthetic line list data for different outbreak scenarios and to real-world data of hospitalizations during the COVID-19 pandemic in Switzerland. We found that under realistic delays between symptom onset and reporting of hospitalized cases, nowcasts for days close to the present are based on few reported cases and thus can be substantially influenced by prior assumptions about the smoothness and shape of the epidemic curve. In nowcasting practice, it is therefore important to be aware that under long reporting delays,

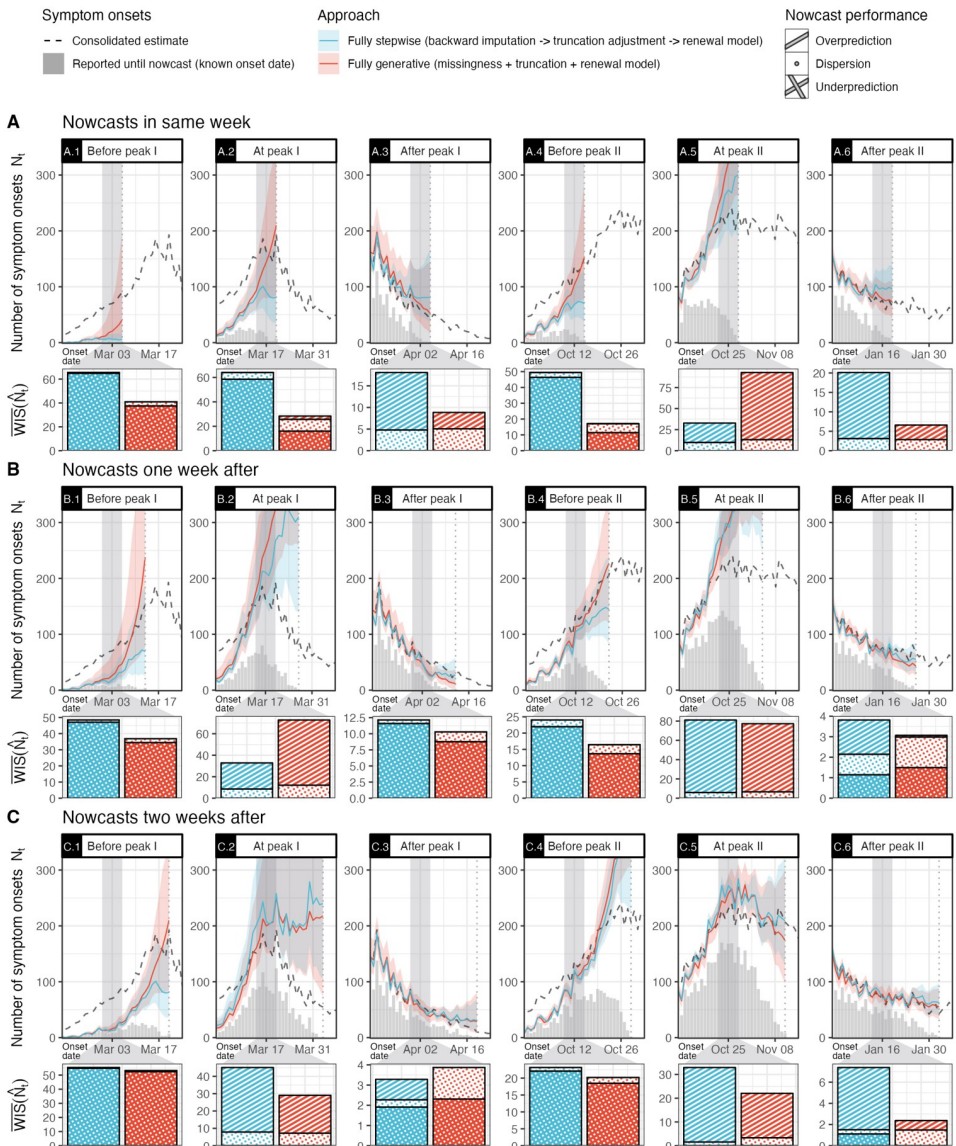

**Fig 6. Nowcasts of $N_t$ on incomplete hospitalization line list data during the COVID-19 pandemic in Switzerland.**
Shown are nowcasts with 95% credible interval (CrI) for the number of hospitalizations with COVID-19 by date of
symptom onset $N_t$ in different phases of the first and second wave, obtained through i) a fully stepwise approach using
cases by date of symptom onset with a backward imputation step and a truncation adjustment step (blue), and ii) a
fully generative approach using an integrated missingness, truncation and renewal model (red). The direct approach
using cases by date of report cannot produce nowcasts of $N_t$. Also shown are the number of cases by symptom onset
date reported until the respective nowcast date (grey bars), and consolidated point estimates (7–14 days after
maximum delay, averaged over all models) of the true $N_t$ (black dashed lines). Shown below each phase is the weighted
interval score (WIS, lower is better) for $N_t$ nowcasts of each approach during a selected week (grey shade) evaluated on
the consolidated point estimates. The scores (colored bars) are decomposed into penalties for underprediction
(crosshatch), dispersion (circles), and overprediction (stripes). Results are shown for nowcasts made at different lags
from the selected week (vertical dotted lines), i. e. at the end of the selected week (top row), one week later (middle
row), and two weeks later (bottom row).

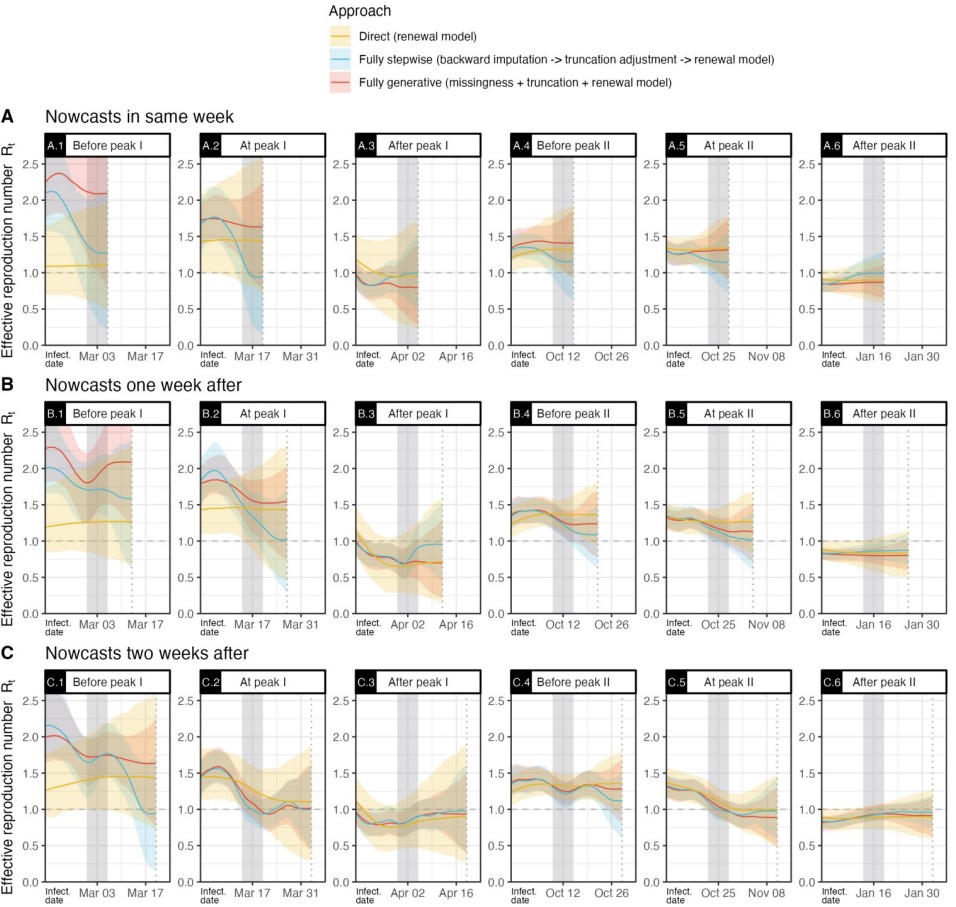

**Fig 7. Nowcasts of $R_t$ on incomplete hospitalization line list data during the COVID-19 pandemic in Switzerland.**
Shown are nowcasts with 95% credible interval (CrI) for the effective reproduction number $R_t$ in different phases of the
first and second wave, obtained from hospitalization line list data through i) a direct approach using cases by date of
report with no truncation adjustment (yellow), ii) a fully stepwise approach using cases by date of symptom onset with
a backward imputation step and a truncation adjustment step (blue), and iii) a fully generative approach using cases by
date of symptom onset with an integrated missingness, truncation, and renewal model (red). Results are shown for
nowcasts made at different lags from the selected week (vertical dotted lines), i. e. at the end of the selected week (top
row), one week later (middle row), and two weeks later (bottom row).

real-time nowcasts can effectively become weakly informed forecasts due to a lack of data
signal.

When nowcasting $R_t$, estimates can be obtained either from case counts by date of report
$C_t$, or from case counts by date of symptom onset $N_t$. The approach using $C_t$ can be more
direct as it requires no truncation adjustment of case counts, but it depends on external esti-
mates of the reporting delay distribution to infer the time series of infections. Our results show
that the resulting $R_t$ estimates can still be substantially biased when using naive empirical esti-
mates of the reporting delay. On the other hand, approaches using cases by date of symptom
onset are more complex as they must adjust for the right truncation of the real-time case
count. Existing methods conduct this truncation adjustment in a separate step, and we found
that this can systematically over- and underpredict case counts depending on the current epi-
demic phase. Specifically, we demonstrated that when modeling the expected number of cases
via a stationary random walk—as customary in earlier studies [15, 17, 18, 35, 38]—nowcasts

will be biased during periods of exponential growth or decline. Notably, our results show that the stationary smoothing can even overwrite prior assumptions in the subsequent $R_t$ estimation step and therefore bias the $R_t$ estimates towards $R_t = 1$. In our proposed generative approach, the renewal model assumed for $R_t$ estimation also serves as an epidemiologically motivated smoothing prior for the case counts. As a result, the generative model extrapolated the current transmission dynamics, avoiding such bias for nowcasts of $N_t$ during phases of exponential growth or decline. However, our renewal model does not predict changes in transmission such as from a depletion of susceptible individuals in the population or due to non-pharmaceutical interventions. Therefore, the generative approach typically overpredicted $N_t$ at the immediate peak of the epidemic curve, and—like the stepwise approach—only detected change points in $R_t$ after a delay of more than one week. In our analysis of data from the COVID-19 pandemic in Switzerland, we found that the limitations of both approaches can be exacerbated in the beginning of an outbreak, when reporting is first established and then accelerates significantly. If such changes in reporting delays are not correctly estimated in real-time, nowcasts for $N_t$ will be susceptible to underprediction during the growth phase and overprediction at the peak.

When nowcasting from incomplete line list data with missing symptom onset dates, we found that even with up to 60% of missing data, informative nowcasts of the total number of symptom onset dates could be made. To account for the missing onset dates, we implemented two stepwise approaches, i.e. with an additional i) independent imputation step and ii) backward-delay imputation step, as well as a generative approach that uses an integrated missingness model. As multiple imputation was not computationally feasible, the stepwise approaches only used a single imputation and thereby ignored uncertainty arising from the missing dates. In contrast, our generative model jointly fits to the complete and incomplete cases in the line list and naturally accounts for uncertainty arising from missing onset dates. Notably, we find that for short nowcasting lags, when few cases have been reported and overall nowcasting uncertainty is high, performance differences between the approaches were mostly negligible. At longer lags, however, the stepwise approach with independent imputation showed bias from misspecified delays especially around the peak of the epidemic curve. In contrast, the stepwise approach with backward-delay imputation correctly modeled delays during imputation and was thus unbiased but overconfident as it did not account for imputation uncertainty. $N_t$ nowcasts using the generative approach with an integrated missingness model had the highest uncertainty and achieved superior performance at longer lags. The resulting performance differences were however mostly with regard to $N_t$, and no clear differences with regard to $R_t$ could be identified, where the overall uncertainty is generally high. In the setting studied here, it thus seems that the uncertainty from missing onset dates may be small compared to the uncertainty from reporting delays and infection dynamics. In our model, we specified $\alpha_t$, the probability for symptom onset dates to be known, with respect to the onset date $t$. It should be noted that in settings where symptom onset information for already reported cases is entered retrospectively in the line list, also known as backfilling, this probability may also depend on the date of report. That is, the probability for symptom onset dates to be known could be systematically lower for cases reported closer to the present. In such a setting, our generative model may still correctly account for cases with missing onset date, however, this could require a different prior for $\alpha_t$ which adequately represents a downward trend towards the present. When backfilling is the main source of missingness, it may also be preferable to index $\alpha_t$ by the date of report instead.

In our models, we used a common but simplistic non-parametric smoothing prior, i. e. a first-order random walk, to smooth the time-varying parameters. In practice, a large variety of smoothing priors can be used for this purpose, with the potential of improving the

performance of each approach. In a supplementary analysis, we found that when the stepwise approach is used with a non-stationary, exponential smoothing prior [53], the tendency to over- or underpredict case counts during phases of epidemic growth or decline is strongly reduced. Nevertheless, some bias remained and it should be noted that the use of more complex smoothing priors may also introduce other, more subtle biases to $R_t$ estimation. At the same time, non-stationary smoothing priors such as innovations state space models [53] or Gaussian processes [2, 54] can also be employed to smooth $R_t$ in a generative model, potentially enabling tighter credible intervals and better capturing changes in transmission dynamics. Moreover, additional knowledge or data could be integrated in such priors, e. g. to provide more information on time-varying reporting delays or to account for population mobility and non-pharmaceutical interventions [55, 56]. In the latter case, a generative approach offers the advantage of modeling population-level effects on disease transmission directly via $R_t$ instead of via the expected number of symptom onsets $\lambda_t$, thereby ensuring temporal accuracy of the effect model. Moreover, due to the joint inference approach, better estimation of reporting delays may also improve the handling of missing onset dates, and more accurate models of transmission dynamics may also improve nowcasts of case counts. However, as the suitability of each option will vary with the specific setting and application, it is important that practical tools for real-time surveillance allow for flexible model specification. In line with the generative approach described in this paper, we have contributed a missingness and renewal model component to the open-source R package *epinowcast* [57].

Overall, our findings highlight the importance of adequate smoothing priors for nowcasting under long delays, and demonstrate how model misspecification can impact downstream estimates in stepwise modeling pipelines. Our results on synthetic data suggest that the various delays (infection to symptom onset, symptom onset to report) in case reporting data can only be partially compensated by nowcasting methods and that changes in transmission dynamics cannot be estimated arbitrarily close to the present. From a public health perspective, this means that nowcasts from case reporting data may only provide robust information for action after a certain time lag, and that nowcasts very close to the present should rather be interpreted as bridging the gap between retrospective estimation and short-term forecasting. The use of such projections can vary with the epidemic situation and policy questions at hand. However, we argue that a generative approach as developed in this work has several generally desirable properties for situational awareness and short-term planning. First, absent a sufficient signal of changes in transmission, a continuation of the recent transmission dynamics can be seen as the preferred minimal assumption [58]. Methods with a bias towards $R_t = 1$ will predict changes in transmission by default and are therefore difficult to interpret. In contrast, our generative approach reflects growth trends during the early phase of an outbreak, and estimates changes in $R_t$ only when indicated by data. Second, the use of Bayesian generative modeling enables accurate and comprehensive uncertainty quantification by default, and does not require decisions about the number of resampling iterations or multiple imputations. Third, by avoiding intermediate, non-parametric smoothing, and combining all steps in a single model, the generative approach is particularly well suited to integrating situational knowledge and additional data on transmission without the risk of introducing inconsistencies between steps.

We note several general limitations of the methods used in this work. First, to model changes in the reporting delay over time, we have used a time-to-event model formulation in which effects by date of onset and date of report are applied proportionally on the logit scale to the hazard for each delay [17]. This assumption reduces model complexity, but limits the ability to represent certain types of distributional shifts over time and excludes negative delays. Second, we treated reported symptom onset dates as exact, while in reality there may be

uncertainty and inconsistency in how onset dates are ascertained. We also note that line lists can include patients with asymptomatic infection, in which case the imputed symptom onset dates are only hypothetical events that serve as a proxy for the date of infection. This may be the case, for example, for patients who are hospitalized for other reasons and become infected in hospital, but will only bias $R_t$ estimates if the proportion of infections in hospital to the total number of infections changes over time. Third, like previous methods, our models encode a missing-at-random assumption for cases with missing symptom onset date. If missingness correlates with reporting delay, this assumption will be violated and could lead to bias in imputation and truncation adjustment. It is therefore recommendable to conduct sensitivity analyses if additional data about the presence of symptoms is available. In theory, such information could also be integrated into our framework to explicitly model correlation between reporting delays and missingness. Fourth, we assumed that the distributions for the generation interval and the incubation period are exactly known, which is seldom the case in practice. However, both in the generative and the stepwise approach, it is possible to account for uncertainty in these parameters [2]. Fifth, we assumed a fixed ascertainment proportion $\rho_t = \rho$ over time to ensure identifiability. Under this assumption, exact knowledge of the ascertainment proportion is not essential because a misspecification of $\rho$ does not bias $R_t$ and distorts uncertainty quantification only in extreme cases. It thus allows us to obtain $R_t$ estimates from hospitalization data that are indicative of population-wide transmission, but will lead to bias in periods where $\rho_t$ changes. This is particularly important when there are differences in reporting delays or ascertainment proportions for cases from distinct subpopulations, e. g. age groups. In such a case, it can be necessary to add group structure to the renewal and reporting delay components of the model [27, 57, 59], to avoid bias from differing transmission dynamics between the subpopulations. Moreover, time-varying ascertainment proportions may also estimated by fitting to several data sources, e. g. hospitalizations and deaths, simultaneously. Sixth, we have only tested a simplistic version of the direct $R_t$ estimation approach, with naive external estimates of the reporting delay. In practice, reporting delays may be externally estimated using more sophisticated methods that account for right truncation and censoring, and the $R_t$ estimation model can account for weekday effects and other modifiers of case counts [26]. Last, in our assessments using synthetic data, we stratified the performance of different approaches by epidemic phase. While this is helpful for understanding the behavior of a model, we do not recommend this approach for model selection in real-time surveillance, as it is generally difficult to identify the current epidemic phase.

We conclude by noting that the generative model proposed in this work naturally encompasses other forms of $R_t$ estimation. For example, in settings where no line list data is available, $R_t$ can only be estimated from the time series of reported cases as described in Section 2.1. However, this scenario corresponds to a special case in the generative model where all symptom onset dates are missing and a strong prior on the reporting delay is used. Similarly, $R_t$ is sometimes only estimated retrospectively, when case data are already fully reported, with some onset dates missing. This corresponds to a case in our model where all days of interest are sufficiently far in the past, beyond the maximum reporting delay. In addition, we have here used the example of nowcasting by date of symptom onset, but our framework is similarly applicable to other dates of reference, such as the date of positive test.

By describing how cases arise from an infection process and are reported with time-varying stochastic delays and partially missing dates, generative modeling offers an interpretable framework for the development of user-friendly yet extensible nowcasting tools. Together with up-to-date, de-identified line list data with information on reporting delays, such tools are essential for real-time surveillance during epidemic outbreaks.

## Supporting information

**S1 Appendix. Supplementary nowcasting results and details on modeling, implementation, and simulation.**
(PDF)

## Acknowledgments

We thank Felix Günther for insightful discussions about the modeling of missing data. We thank the Federal Office of Public Health (FOPH) in Switzerland for their collaboration and the provision of COVID-19 line list data for this study.

## Declarations

### Ethics approval

Ethics approval was not required for this study.

## Author Contributions

**Conceptualization:** Adrian Lison, Tanja Stadler.

**Data curation:** Adrian Lison.

**Formal analysis:** Adrian Lison.

**Funding acquisition:** Tanja Stadler.

**Investigation:** Adrian Lison.

**Methodology:** Adrian Lison.

**Project administration:** Adrian Lison.

**Resources:** Tanja Stadler.

**Software:** Adrian Lison, Sam Abbott.

**Supervision:** Tanja Stadler.

**Validation:** Adrian Lison, Jana Huisman.

**Visualization:** Adrian Lison.

**Writing – original draft:** Adrian Lison.

**Writing – review & editing:** Adrian Lison, Sam Abbott, Jana Huisman, Tanja Stadler.

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
