## [Decision Letter · Decision Letter 0]

11 Jan 2024

Dear Mr. Lison,

Thank you very much for submitting your manuscript "Generative Bayesian modeling to nowcast the effective reproduction number from line list data with missing symptom onset dates" for consideration at PLOS Computational Biology.

As with all papers reviewed by the journal, your manuscript was reviewed by members of the editorial board and by several independent reviewers. In light of the reviews (below this email), we would like to invite the resubmission of a significantly-revised version that takes into account the reviewers' comments.

First I would like to appologize for the long handling time. It is explained by unusual hard problems finding two referees. In fact, the second referee was found only after asking 14 tentative referees ...

Any, the paper has been read by two experts in the field, and less detailed by myself. All of us find the manuscript interesting. However, both referees suggests several vchanges before being potentially acceptable for publication in PCB. In particular, referee 2 has some thoughts on improving applicability of the method. Please revise the manuscript following comments by both referees.

Kind regards, Tom Britton, Associate editor

We cannot make any decision about publication until we have seen the revised manuscript and your response to the reviewers' comments. Your revised manuscript is also likely to be sent to reviewers for further evaluation.

Sincerely,

Tom Britton

Academic Editor

PLOS Computational Biology

Thomas Leitner

Section Editor

PLOS Computational Biology

First I would like to appologize for the long handling time. It is explained by unusual hard problems finding two referees. In fact, the second referee was found only after asking 14 tentative referees ...

Any, the paper has been read by two experts in the field, and less detailed by myself. All of us find the manuscript interesting. However, both referees suggests several vchanges before being potentially acceptable for publication in PCB. In particular, referee 2 has some thoughts on improving applicability of the method. Please revise the manuscript following comments by both referees.

Kind regards, Tom Britton, Associate editor

Reviewer's Responses to Questions

**Comments to the Authors:**

Reviewer #1: The review is uploaded as an attachment.

Reviewer #2: In this manuscript Lison et al. propose a method to estimate the instantaneous effective reproductive number from surveillance line-list data of COVID-19 using a bayesian framework and MCMC based on daily incidence over time. The method aims to solve two major problems in communicable disease near-real time surveillance, namely missingness and right truncation, together with estimation of R_eff. The integrative method is proposed as an alternative to stepwise approaches, aiming to improve the propagation of uncertainty. They further test their method using in silico and empirical data and compare its performance with a representative stepwise approach. The proposed method requires prior estimates of the serial interval/incubation period, unlike other recently proposed methods (Dai et al. 2023). The study is written in a clear manner and the authors explain well the formulated model and the validation steps. Also, the authors provide an R package extension for implementation for real-time surveillance.

There are some points that might need to be addressed, particularly concerning the limitations of the method when used in public health surveillance. In the following, this reviewer provides with a series of comments that mainly focus on the potential usefulness of the approach:

Line 95. This is a key assumption (constant ascertainment rates) which is likely not true most of the time in reality, particularly in the first stage of a pandemic/outbreak. Authors could test the performance of the methods (integrated vs. stepwise) under violations of this assumption. Otherwise, as this is a limitation that is well known but not easily addressed with statistical approaches, it would be reasonable to highlight it within the limitations in the discussion (although it is well remarked in the methods, later line 102).

Line 97 assuming a prior incubation period, although obviously useful for applying the method, makes the model less useful for outbreaks of emerging pathogens (as they are typically not well understood in the early stages). Would be interesting to see how deviations from this assumption (i.e. misspecified incubation period) affects nowcasting.

Line 183. This might be an important claim for practical implementation although it is difficult to understand how this might play a role in surveillance systems. Can the authors provide more details on how this cost can impact routine surveillance?

Lines 213-244. Congratulations for this paragraph. This is a commonly misunderstood problem that is well explained here.

Can authors provide more details on the pooled overdispersion parameterization and robustness of the approach? Model results might be sensitive to the prior here.

Line 306. This seems like a strong assumption with no reference/clear justification

Line 292. It is unclear whether the data is simulated, analyzed and nowcasted daily or weekly, which is a very important aspect. Also the results plots (Figure 1-2) misses the y-axis label (e.g., Day by date of symptoms onset).

Line 327-344. Missing reference/s for the WIS approach.

Estimation of transmission variables from hospitalization records can be severely biased, particularly in the beginning of a pandemic and across different populations (Sherratt et al. 2021). Lines 371-376 could be brought up earlier in the subsection for clarity.

While it is well detailed, it is difficult to follow the description of the differences between the approaches in the text, having to go back-and-forth to the figures. For clarity, I would recommend producing a table where the main, overall results (such as over- or under-estimation at different time-steps) are summarized.

While formal comparison (based on WIS) shows better performance of the generative method in general, when applying the method in real world public health surveillance would require acknowledging the limitations of its performance. For example, while in Figure 2 first column row B and C, the generative method performs better, the nowcasted trend can be as misleading as that nowcasted by the stepwise approach from a public health perspective. This reviewers’ interpretation of the overall results is that both methods somehow fail to provide reliable information, or at least, that the information that can be obtained in the very beginning of an outbreak with limited data is almost not useful for predicting transmission dynamics. Similar general interpretation can be made for R_eff estimates and when including missingness (Figures 3 , 4, 5). On the other hand, with downward trends and more data, the performance of both methods is reasonably good once they control for well known bias.

The issues raised in the previous point also applied for the hospitalization data, with even more extreme trends (exponential trend nowcasted by generative method, over estimation etc)

Line 565:. This somehow reflects the previous points. While the claim that the generative model performs statistically better in epidemic growth or decline is based on the WIS is true, from a surveillance perspective this might not necessarily be true, as the information that public health might be looking for (e.g., confirmation that the epidemic curve is stalling due to interventions) can not necessarily be obtained from the model nowcasts. Thus, while the method deals better with uncertainty and likely has better computational cost, the claim of a better performance, at least from a public health perspective (i.e., which information is useful) is arguable. Acknowledging the limitations of the generative method for its use in public health, not only from an statistical point of view, as acknowledging that of stepwise approaches, can improve the discussion on which methods are more useful and when can we use them reliably.

References

Dai, Chenxi, Dongsheng Zhou, Bo Gao, and Kaifa Wang. 2023. “A New Method for the Joint Estimation of Instantaneous Reproductive Number and Serial Interval during Epidemics.” PLoS Computational Biology 19 (3): e1011021.

Sherratt, Katharine, Sam Abbott, Sophie R. Meakin, Joel Hellewell, James D. Munday, Nikos Bosse, CMMID COVID-19 Working Group, Mark Jit, and Sebastian Funk. 2021. “Exploring Surveillance Data Biases When Estimating the Reproduction Number: With Insights into Subpopulation Transmission of COVID-19 in England.” Philosophical Transactions of the Royal Society of London. Series B, Biological Sciences 376 (1829): 20200283.

**Have the authors made all data and (if applicable) computational code underlying the findings in their manuscript fully available?**

Reviewer #1: Yes

Reviewer #2: Yes

PLOS authors have the option to publish the peer review history of their article (what does this mean?). If published, this will include your full peer review and any attached files.

Reviewer #1: **Yes: **Fanny Bergström

Reviewer #2: No
---

## [Decision Letter · Decision Letter 1]

22 Mar 2024

Dear Mr. Lison,

We are pleased to inform you that your manuscript 'Generative Bayesian modeling to nowcast the effective reproduction number from line list data with missing symptom onset dates' has been provisionally accepted for publication in PLOS Computational Biology.

Best regards,

Tom Britton

Academic Editor

PLOS Computational Biology

Thomas Leitner

Section Editor

PLOS Computational Biology

Associate editor

Both referees are happy with the revision and so am I. I therefore recommend that the paper is accepted for publication.

Kind regards, Tom Britton

Reviewer's Responses to Questions

**Comments to the Authors:**

Reviewer #1: Thank you for your thourough revision and consideration of my feedback. I am pleased with all changes made.

Reviewer #2: The authors have now addressed thoroughly the comments and queries, and properly acknowledge the limitations of the model. I believe the manuscript is clear and rigorous, and will contribute positively to improving nowcasting of infectious diseases from surveillance data, highlighting the new methodological approaches and gaps that are faced. I therefor recommend the manuscript for publication without further comments.

**Have the authors made all data and (if applicable) computational code underlying the findings in their manuscript fully available?**

Reviewer #1: Yes

Reviewer #2: Yes

PLOS authors have the option to publish the peer review history of their article (what does this mean?). If published, this will include your full peer review and any attached files.

Reviewer #1: **Yes: **Fanny Bergström

Reviewer #2: No

---

## [Editor Report · Acceptance letter]

6 Apr 2024

PCOMPBIOL-D-23-01366R1 

Generative Bayesian modeling to nowcast the effective reproduction number from line list data with missing symptom onset dates

Dear Dr Lison,

I am pleased to inform you that your manuscript has been formally accepted for publication in PLOS Computational Biology. Your manuscript is now with our production department and you will be notified of the publication date in due course.

With kind regards,

Zsofia Freund
